# Riemannian High-Order Pooling for Brain Foundation Models

**Chen Hu**[1,2]*, **Ziheng Chen**[3]*, **Rui Wang**[1]†, **Yefeng Zheng**[2], **Nicu Sebe**[3]

[1] School of Artificial Intelligence and Computer Science, Jiangnan University
[2] Medical Artificial Intelligence Lab, Westlake University
[3] Department of Information Engineering and Computer Science, University of Trento

## ABSTRACT

Electroencephalography (EEG) is a noninvasive technique for measuring brain electrical activity that supports a wide range of brain-computer interaction applications. Motivated by the breakthroughs of Large Language Models (LLMs), recent efforts have begun to explore Large EEG foundation Models trained on broad unlabeled corpora. However, most advances focus on improving the backbone while neglecting the classification head. Existing models often rely on a single class token, underutilizing the spatiotemporal structure and second-order statistics that are crucial for EEG decoding. We propose Riemannian High Order Pooling (RHOP), a plug-and-play module that injects principled Riemannian statistics into the classifier. RHOP maps each token to a quotient Gaussian jointly encoding mean and second-order information, yielding scale-invariant descriptors. Tokens are then aggregated by estimating a Riemannian Gaussian on the SPD manifold, where the Fréchet mean and covariance are embedded into an SPD descriptor. The resulting normalized vector is fused with the class token for prediction. RHOP is backbone-agnostic and integrates with modern EEG foundation models, *e.g.*, BIOT and LaBraM. Across diverse EEG benchmarks, it improves accuracy and efficiency under full fine-tuning, linear probing, and from-scratch training settings. The code is publicly available at github.com/ChenHu-ML/RHOP.

## 1 INTRODUCTION

Electroencephalography (EEG), which records cortical electrical potentials with millisecond precision, provides dynamic insights into brain function. It has enabled advancements in seizure detection (Ahmad et al., 2022; Cherian & Kanaga, 2022), sleep staging (Aboalayon et al., 2016; Phan & Mikkelsen, 2022; Zhou et al., 2025), motor imagery (Altaheri et al., 2023; Ju & Guan, 2023; Roy et al., 2019), abnormality screening (Roy et al., 2019), emotion analysis (Suhaimi et al., 2020; Biesmans et al., 2016; Dadebayev et al., 2022), and auditory attention (Biesmans et al., 2016). However, EEG's practical deployment remains challenging due to issues like low signal-to-noise ratio, inter-subject variability, and task-dependent non-stationarity (Hine et al., 2017).

Early EEG decoding pipelines relied on traditional machine-learning methods (Lotte et al., 2007). With the rise of deep learning (LeCun et al., 2015; Zhao et al., 2024; Liu et al., 2024; Gao et al., 2025b;a; Hu et al., 2025b), classic architectures such as Convolutional Neural Networks (CNNs) (Lawhern et al., 2018) and Long Short-Term Memory (LSTM) networks (Phan et al., 2019) were adapted to EEG tasks, followed by transformer backbones (Peh et al., 2022b). In parallel, geometric learning approaches have gained traction in EEG decoding tasks, most notably those leveraging Riemannian geometry. The power and spatial distribution of multi-channel EEG segments can be encoded into covariance matrices, which are symmetric positive definite (SPD) matrices. By operating on the SPD manifold, Riemannian methods exploit metrics that are robust to outliers and noise (Congedo et al., 2017), leading to broad success in practice (Pan et al., 2022; Kobler et al., 2022; Ju et al., 2024; Li et al., 2025). More recently, inspired by the rise of self-supervision and foundation models in vision and language, research has followed this paradigm (Devlin et al.,

---

*Equal contribution.
†Corresponding author: cs_wr@jiangnan.edu.cn

2018; Radford et al., 2021; Bommasani et al., 2021). EEG foundation models are pretrained on large unlabeled corpora with contrastive learning, masked reconstruction, or self-prediction and then transferred to diverse downstream tasks (Yang et al., 2023; Jiang et al., 2024; 2025).

Despite these advances, many foundation models still apply Global Average Pooling (GAP) or concatenate tokens before the final classification, which discards valuable second-order information and underuses global spatiotemporal dependencies. Intuitively, EEG features exhibit dependencies inherently across temporal and channel dimensions (Song et al., 2021a; Wang et al., 2025a). Global Covariance Pooling (GCP) replaces GAP by summarizing activations with a covariance descriptor (Lin et al., 2015; Wang et al., 2017), partially closing this gap. However, typical GCP compresses all tokens into a single covariance matrix, thereby overlooking the intrinsic spatiotemporal structure of EEG features (Lin et al., 2015; Li et al., 2017b; 2018). This raises a central question for applying EEG foundation models to downstream decoding: Can we design a global pooling head that is both statistics-aware and geometry-aware while respecting the underlying spatiotemporal structure?

We answer this question with Riemannian High-Order Pooling (RHOP), a plug-and-play module for EEG foundation backbones and, to our knowledge, the first geometric pooling head tailored for this setting. RHOP is motivated by two empirical properties of EEG features: significant spatiotemporal structure and pervasive scale variation across temporal segments. First, we introduce a quotient-Gaussian embedding that normalizes per-token covariances to correlation form and jointly encodes first- and second-order statistics, removing temporal scale discrepancies while preserving dependency structure (Lovrić et al., 2000a; Thanwerdas & Pennec, 2022). Second, we aggregate information across tokens by estimating a Riemannian Gaussian on SPD manifolds and embedding it into an SPD descriptor, efficiently capturing high-order interactions (Pennec, 2006). Finally, a sparse inverse-covariance layer emphasizes partial correlations and yields a compact vector, which is fused with the classification (CLS) token for prediction (Rahman et al., 2023). RHOP is architecture-agnostic and can be attached to modern EEG backbones such as BIOT and LaBraM (Yang et al., 2023; Jiang et al., 2024). RHOP bridges EEG foundation models with Riemannian statistics by embedding token-level representations into SPD manifolds and preserving their spatiotemporal information and high-order dependencies in a pooling head. In summary, our contributions are threefold:

- **Quotient-Gaussian embedding.** A scale-invariant embedding that transforms per-token covariances into correlation form and jointly encodes first- and second-order statistics, addressing variance differences across temporal segments.
- **Riemannian High-Order Pooling.** A geometry-aware pooling head that preserves token-level spatiotemporal structure and captures high-order interactions via a Riemannian Gaussian-embedded SPD descriptor.
- **Comprehensive empirical validation.** RHOP delivers consistent gains and robust generalization across EEG decoding tasks under state-of-the-art EEG foundation backbones (BIOT and LaBraM) and three training regimes (full fine-tuning, linear probing, and training from scratch).

## 2 RELATED WORKS

**SPD manifold-based EEG decoding.** Methods that operate on the SPD manifold achieve strong EEG decoding by respecting covariance geometry and improving robustness across subjects (Congedo et al., 2017; Ju et al., 2025). For example, Pan et al. (2022); Wang et al. (2025b) built manifold attention directly on SPD manifolds and captures spatiotemporal dependencies, outperforming deep baselines. SPDDSMBN (Kobler et al., 2022) learns domain-invariant tangent-space mappings for unsupervised adaptation with an interpretable normalization scheme. DGCCA (Ju et al., 2024) introduces geodesic correlation with an SPD latent space to align paired covariance modalities. SPDIM (Li et al., 2025) addresses source-free adaptation with conditional and label shift through an SPD-constrained parameterization. Taken together, these works show that geometry-aware learning on SPD manifolds is powerful and motivate pairing such geometric bias with large-scale pretraining.

**Brain foundation models.** Foundation models are large self-supervised systems trained on broad data and adapted to many tasks via fine-tuning (Bommasani et al., 2021), with BERT (Devlin et al., 2018), CLIP (Radford et al., 2021), and GPT-4 (OpenAI, 2023). This paradigm is extending to brain signals. Recent EEG studies pretrain with contrastive learning, masked reconstruction, or self-prediction to learn transferable representations (Banville et al., 2021; Kostas et al., 2021; Chien et al., 2022; Wang et al., 2023; Zhang et al., 2023; Mohammadi Foumani et al., 2024). BIOT learns generic biosignal representations for joint pretraining and cross-dataset transfer (Yang et al., 2023), LaBraM

predicts masked neural tokens for general EEG features (Jiang et al., 2024), CBraMod decouples spatial and temporal modeling with a criss-cross transformer and masked reconstruction for strong cross-dataset generalization (Wang et al., 2025a), and NeuroLM tokenizes EEG signals and uses an LLM with multi-channel autoregression and instruction tuning to unify diverse tasks (Jiang et al., 2025). Despite progress, many backbones compress EEG features into a single CLS token or concatenate tokens before the final classification, leaving the global spatiotemporal structure underused, which motivates our geometry-aware high-order pooling head.

**Gaussian embedding.** Gaussian embedding offers a geometric route to compare distributions. The Fisher–Rao metric formalizes information geometry (Rao, 1945), but closed-form geodesics are challenging for multivariate Gaussians. A practical remedy identifies each Gaussian with an SPD matrix by viewing the Gaussian family as a Riemannian symmetric space, which enables affine-invariant tools for learning (Lovrić et al., 2000a; Pennec, 2006). Alternative lines embed Gaussians into the Siegel domain to define distances (Calvo & Oller, 1990), or relate Gaussians to affine matrix subspaces (Gong et al., 2009). Log–Euclidean mappings linearize SPD structure and yield efficient Gaussian descriptors for vision (Arsigny et al., 2005a; Li et al., 2017a). Deep architectures have adopted these ideas, for example, a global Gaussian layer that maps images to SPD features (Wang et al., 2017) and a Lie-group embedding of Riemannian Gaussians for sets of SPD matrices (Nguyen, 2021). This geometric toolbox naturally connects to second-order pooling.

**Global covariance pooling.** GCP replaces global average pooling with second-order covariance descriptors (Lin et al., 2015; Wang et al., 2017), which can be viewed as modeling feature activations with Gaussian statistics. Early works mainly focused on numerical stabilization of covariance matrices through matrix logarithm backpropagation (Ionescu et al., 2015), matrix power normalization such as MPN-COV (Li et al., 2017b), and iterative matrix square root as in iSQRT-COV (Li et al., 2018). Subsequent methods improved conditioning and scalability, introduced graph-aware formulations (Rahman et al., 2020; Zhu et al., 2024), and enhanced discriminability via sparse inverse covariance estimation (SICE) (Rahman et al., 2023). More recently, matrix functions in GCP have been interpreted from a Riemannian perspective, relating covariance transformations to non-Euclidean classifiers (Chen et al., 2024c;a; 2025). Despite these advances, standard GCP on EEG flattens all spatiotemporal tokens into a single feature dimension before covariance estimation, thereby overlooking intrinsic temporal structure. This limitation motivates our Riemannian high-order pooling module.

## 3 PRELIMINARIES

This section provides a brief overview of the foundations of SPD geometry and the concept of Fréchet Mean (FM). For detailed discussions, please refer to Pennec (2006); Arsigny et al. (2005b).

**SPD manifold.** The space of $n \times n$ SPD matrices is denoted as $\mathcal{S}_n^+ = \{P \in \mathbb{R}^{n \times n} \mid P = P^\top, P \succ 0\}$. It forms a Riemannian manifold when endowed with the Affine-Invariant Metric (AIM) (Pennec, 2006). For two points $P, Q \in \mathcal{S}_n^+$, the AIM distance is defined as

$$d_{\mathrm{AIM}}(P, Q) = \|\log(P^{-\frac{1}{2}} Q P^{-\frac{1}{2}})\|_F, \tag{1}$$

where $\|\cdot\|_F$ is the Frobenius norm and $\log(\cdot)$ denotes the matrix logarithm. The associated exponential and logarithm maps are given by

$$\mathrm{Exp}_P(S) = P^{\frac{1}{2}} \exp(P^{-\frac{1}{2}} S P^{-\frac{1}{2}}) P^{\frac{1}{2}}, \quad S \in T_P \mathcal{S}_n^+, \tag{2}$$

$$\mathrm{Log}_P(Q) = P^{\frac{1}{2}} \log(P^{-\frac{1}{2}} Q P^{-\frac{1}{2}}) P^{\frac{1}{2}}, \quad Q \in \mathcal{S}_n^+, \tag{3}$$

where $\exp(\cdot)$ denotes the matrix exponential, and $T_P \mathcal{S}_n^+$ is the tangent space at $P$.

**Fréchet Mean.** Given a set of points $\{P_i\}_{i=1}^N \subset \mathcal{M}$, the FM is the point $S$ that minimizes the average sum of squared distances to all points. The FM (Karcher, 1977b) is defined as

$$\mathrm{FM}(\{P_i\}) = \underset{S \in \mathcal{M}}{\arg\min} \frac{1}{N} \sum_{i=1}^N d(P_i, S)^2, \tag{4}$$

where $d(P_i, S)$ is the distance between them. The FM is locally unique on general manifolds (Afsari, 2011). In the case of $(\mathcal{S}_n^+, d_{\mathrm{AIM}})$, it is globally well-defined and unique (Chakraborty et al., 2022).

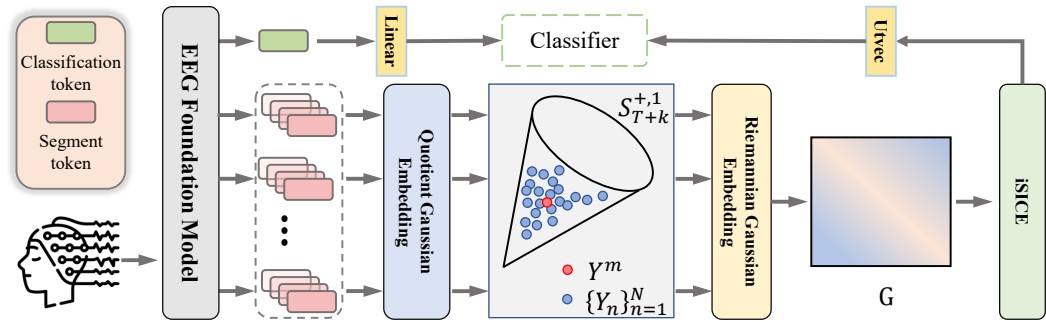

Figure 1: Overview of our RHOP framework. The EEG backbone outputs a CLS token and token-wise features. Each token is transformed into a quotient Gaussian and embedded as $Y_n \in \mathcal{S}_{T+k}^{+,1}$ on the SPD manifold. The set $\{Y_n\}$ is then aggregated into a Riemannian Gaussian, whose FM $Y^m$ and covariance are jointly embedded into an SPD descriptor $G$. Finally, a iSICE + utvec block produces a sparse precision vector, which is fused with the CLS branch for classification.

## 4 RIEMANNIAN HIGH-ORDER POOLING

In this section, we introduce Riemannian High-order Pooling (RHOP), a plug-and-play geometric pooling head for EEG foundation models. The overall RHOP framework consists of three parts: quotient Gaussian embedding, Riemannian Gaussian embedding, and a sparse inverse covariance estimation (iSICE) module (Rahman et al., 2023).

### 4.1 QUOTIENT GAUSSIAN EMBEDDING

Let $(\Sigma, \mu) \in \mathcal{N}(n)$ denote a Gaussian with covariance $\Sigma$ and mean $\mu$. Previous methods often used raw covariance matrices as EEG descriptors (Pan et al., 2022; Li et al., 2025), capturing second-order temporal dependencies but highly sensitive to scale variations. Two tokens with similar temporal dynamics may yield substantially different covariance matrices if their amplitudes differ (Hu et al., 2025a). To address this, we introduce quotient Gaussian distributions, which normalize temporal covariances to eliminate scale dependence. This ensures that the representation focuses on correlation rather than raw magnitudes, thereby providing a scale-invariant descriptor of temporal dynamics.

**Definition 4.1 (Quotient Gaussian Distributions).** Let $(\Sigma, \mu) \in \mathcal{N}(n)$ be a Gaussian distribution. The Quotient Gaussian distribution $\mathcal{QN}(n)$ is defined as

$$\mathcal{QN}(n) \cong \mathcal{N}(n)/\mathrm{Diag}^+(n) = \left\{ [\Sigma, \mu] := \left\{ (\mathcal{D}^{-\frac{1}{2}} \Sigma \mathcal{D}^{-\frac{1}{2}}, \mu) \mid \mathcal{D} \in \mathrm{Diag}^+(n) \right\} \right\}. \quad (5)$$

For the quotient Gaussian $\mathcal{QN}(n)$, each element is an equivalence class of Gaussians $[\Sigma, \mu]$ sharing the same mean and whose covariances differ only by positive diagonal scalings. Within each class, the canonical representative $C = \mathrm{diag}(\Sigma)^{-\frac{1}{2}} \Sigma \mathrm{diag}(\Sigma)^{-\frac{1}{2}}$, is exactly the correlation matrix, which is invariant to scaling. We thus denote each quotient element by $(C, \mu)$, where $C$ is the normalized covariance, *i.e.*, a correlation matrix. Then, to facilitate end-to-end optimization, we embed the quotient Gaussian into SPD matrices as shown in Thm. 4.2 (see App. I for the proof).

**Theorem 4.2 (Quotient Gaussian Embedding ).** *Let $\mathcal{S}_{n+k}^{+,1}$ be the space of SPD $(n + k) \times (n + k)$ matrices with determinant 1. A quotient Gaussian $(C, \mu) \in \mathcal{QN}(n)$ can be identified as*

$$(\det C)^{-\frac{1}{n+k}} \begin{bmatrix} C + k\mu\mu^\top & \mu^{(k)} \\ \mu^{(k)\top} & I_k \end{bmatrix} \in \mathcal{S}_{n+k}^{+,1}, \quad (6)$$

*where $I_k$ is the $k \times k$ identity, and $\mu^{(k)}$ repeats $\mu$ across $k$ identical columns.*

This embedding maps quotient Gaussians into the SPD manifold, enabling the joint representation of mean and normalized covariance within a unified form. Following Nguyen (2021); Lovrić et al. (2000b), we endow this space with the AIM introduced in Sec. 3.

---

**Algorithm 1:** RHOP over SPD manifolds

---

**Input:** EEG features $X \in \mathbb{R}^{D \times T \times N}$ from $f_\theta$; CLS token $y_0$
**Output:** Class probabilities $p$
**for** $n \leftarrow 1$ **to** $N$ **do**
    Compute temporal statistics $(\mu_n, \Sigma_n)$ across channels;
    $C_n \leftarrow \mathrm{diag}(\Sigma_n)^{-1/2} \Sigma_n \, \mathrm{diag}(\Sigma_n)^{-1/2}$;
    $Y_n \leftarrow$ quotient-Gaussian embedding via Eq. (11);
**end**
$Y^m \leftarrow \mathrm{FM}(\{Y_n\}_{n=1}^N)$;    $Y^c \leftarrow$ covariance as in Eq. (12);
$G \leftarrow \mathrm{embed}\,(Y^m, Y^c)$ using Eq. (9);
$g \leftarrow \mathrm{utvec}(\mathrm{SICE}(G; \lambda_{\mathrm{SICE}}))$ ;
$p \leftarrow \mathrm{softmax}\big(\mathrm{FC}([y_0; g])\big)$;
**return** $p$;

---

## 4.2 RIEMANNIAN GAUSSIAN EMBEDDING

Motivated by Euclidean covariance pooling, we extend first- and second-order statistics on $\mathcal{S}_n^+$ via Riemannian Gaussians. Given $\{P_i\}_{i=1}^N \in \mathcal{S}_n^+$, the FM (Fréchet, 1948) is given as

$$P^m = \mathrm{FM}(\{P_i\}_{i=1}^N) = \arg\min_{Y \in \mathcal{S}_n^+} \frac{1}{N} \sum_{i=1}^N d(Y, P_i)^2, \tag{7}$$

where $d_{\mathrm{AIM}}(\cdot, \cdot)$ is the AIM distance defined in Eq. (1). It can be computed via the Karcher flow (Karcher, 1977a), which iteratively maps $P_i$ to $T_P \mathcal{S}_n^+$, calculates the mean in the tangent space, and maps back onto $\mathcal{S}_n^+$ until convergence (Moakher, 2005). Due to the computational cost of FM, we follow the previous work (Brooks et al., 2019; Chen et al., 2024b) and set the number of iterations to one. The implementation steps are provided in App. F.

Similarly, given FM $P^m$, the covariance of $\{P_i\}_{i=1}^N \in \mathcal{S}_n^+$ is defined as

$$P^c = \frac{1}{N-1} \sum_{i=1}^N f_v \left(\mathrm{Log}_{P^m}(P_i)\right) f_v \left(\mathrm{Log}_{P^m}(P_i)\right)^\top, \tag{8}$$

where $f_v(\cdot)$ vectorizes the lower-triangular entries of a symmetric matrix with off-diagonal terms scaled by $\sqrt{2}$ (Pennec et al., 2006). Then, a Riemannian Gaussian is parameterized by the pair $(P^m, P^c) \in \mathcal{S}_n^+ \times \mathcal{S}_{n'}^+$. As shown by Nguyen (2021), this pair lies on a product SPD manifold that forms a Lie group, and the following block-matrix construction provides a Lie–group–isomorphic embedding that preserves its algebraic and geometric structure. Let $P^c = LL^\top$ denote the Cholesky decomposition. The embedding is formulated as

$$(P^m, P^c) \mapsto \begin{bmatrix} L & 0_{n' \times k'} \\ \varphi^{k'}(P^m) & I_{k'} \end{bmatrix} \in \mathcal{S}_{n'+k}^{+,1}, \tag{9}$$

where $I_{k'}$ is a $k' \times k'$ identity matrix and $0_{n' \times k'}$ is a zero block. Here $\varphi$ is chosen as $\varphi = f_v \circ \log$, with $\log(P) = U \log(Z) U^\top$ representing the matrix logarithm by eigenvalue decomposition of $P$.

## 4.3 THE OVERALL FRAMEWORK OF RIEMANNIAN HIGH-ORDER POOLING

In this section, we detail the whole framework of RHOP. An EEG foundation backbone $f_\theta$ (*e.g.*, BIOT or LaBraM) first extracts spatiotemporal features $X \in \mathbb{R}^{D \times T \times N}$, where $D$ is the number of channels, $T$ the number of temporal segments, and $N$ the token length. For convenience we permute to $\widetilde{X} \in \mathbb{R}^{N \times T \times D}$. We then compute temporal first- and second-order statistics for each token $n$,

$$\mu_n = \frac{1}{D} \sum_{i=1}^D \widetilde{X}_{n,:,i} \in \mathbb{R}^T, \quad \Sigma_n = \frac{1}{D-1} \sum_{i=1}^D \left(\widetilde{X}_{n,:,i} - \mu_n\right)\left(\widetilde{X}_{n,:,i} - \mu_n\right)^\top \in \mathbb{R}^{T \times T}. \tag{10}$$

To ensure that $\Sigma_n$ is an SPD matrix, we stabilize it by adding a small multiple of the identity matrix $I$, *i.e.*, $\Sigma_n \leftarrow \Sigma_n + \sigma I$, where we set $\sigma = 0.001$ in all experiments. Let $\mathcal{D}_n = \mathrm{diag}(\Sigma_n)$ and

$C_n = \mathcal{D}_n^{-1/2} \Sigma_n \mathcal{D}_n^{-1/2}$ be the correlation matrix which removes per-time scaling. According to Thm. 4.2, each token is then encoded as a quotient Gaussian embedding, given below

$$Y_n = (\det C_n)^{-\frac{1}{T+k}} \begin{bmatrix} C_n + k\,\mu_n\mu_n^\top & \mu_n^{(k)} \\ \mu_n^{(k)\top} & I_k \end{bmatrix} \in \mathcal{S}_{T+k}^{+,1}. \tag{11}$$

To aggregate information across tokens, we estimate a Riemannian Gaussian on the SPD manifold based on the set $\{Y_n\}_{n=1}^N$. The FM and empirical Riemannian covariance are given by

$$Y^m = \text{FM}\big(\{Y_n\}_{n=1}^N\big), \quad Y^c = \frac{1}{N-1}\sum_{n=1}^N f_v\big(\text{Log}_{Y^m}(Y_n)\big)\,f_v\big(\text{Log}_{Y^m}(Y_n)\big)^\top. \tag{12}$$

The obtained product manifold $(Y^m, Y^c)$ is embedded into an SPD matrix $G$ using Eq. (9).

Finally, RHOP integrates global semantics with statistical structure to enhance prediction. The backbone outputs a CLS token $y_0 \in \mathbb{R}^L$ for global semantics, while the statistical branch applies iSICE (Rahman et al., 2023) to $G$, followed by upper-triangular extraction and vectorization, *i.e.*,

$$g = \text{utvec}\big(\text{iSICE}(G)\big). \tag{13}$$

The final prediction is obtained by concatenating $y_0$ and $g$ and passing them through a linear layer with softmax activation. This design enriches the CLS token with quotient- and Riemannian-based high-order statistics, yielding a more discriminative representation.

## 5 EXPERIMENTS

This section evaluates RHOP on four EEG benchmarks under training from scratch, full fine-tuning, and linear-head tuning settings. We compare our method against state-of-the-art foundation models and representative GCP heads.

### 5.1 EXPERIMENT SETUP

**Datasets.** We evaluate RHOP on four EEG benchmarks spanning abnormal detection, event classification, motor imagery, and event-related potentials. **TUAB** (Obeid & Picone, 2016) contains 23-channel EEG at 256 Hz labeled as normal or abnormal, with 409,455 ten-second segments. **TUEV** (Obeid & Picone, 2016) includes 112,491 five-second segments from 23 channels at 256 Hz across six classes: spike & sharp wave (SPSW), generalized periodic epileptiform discharges (GPED), periodic lateralized epileptiform discharges (PLED), eye movement (EYEM), artifact (ARTF), and background (BCKG). **BCIC2B** (Steyrl et al., 2016) records EEG from 10 subjects on three bipolar channels (C3, Cz, C4) at 250 Hz, with two-class motor imagery (left vs. right hand). Each subject has two no-feedback sessions (120 balanced trials each) and three feedback sessions. **PhysioP300** (Goldberger et al., 2000) is a Donchin-style $6{\times}6$ row/column speller, where rows/columns are flashed for 100 ms with 50 ms inter-stimulus intervals ($\approx$20 flashes each), and each subject spells 20 characters per run.

**Backbones.** We evaluate RHOP on two widely used EEG foundation models: BIOT (Yang et al., 2023) and LaBraM (Jiang et al., 2024). Since public checkpoints are only available for *LaBraM-Base*, we fine-tune this configuration in all experiments. To further assess robustness beyond foundation pretraining, we also train BIOT from scratch without any pretraining.

**Preprocessing.** For TUAB and TUEV, we strictly follow the original backbone preprocessing without any additional modifications. For BCIC2B, we use uniform units, apply a 0–38 Hz band-pass filter, resample the data to 200 Hz, and perform EA normalization (He & Wu, 2019) within each session. For PhysioP300, we also use uniform units, apply a 120 Hz low-pass filter, downsample to 200 Hz, and extract 2 s epochs starting at $-0.7$ s relative to the onset of the flicker stimulus. For more details, please refer to App. C.

**Baselines.** We compare RHOP with both non-foundation and foundation baselines. For non-foundation baselines, we include widely used EEG models such as EEGNet (Lawhern et al., 2018), EEGConformer (Song et al., 2022), SPaRCNet (Jing et al., 2023), ContraWR (Yang et al., 2021), CNN-Transformer (Peh et al., 2022a), FFCL (Li et al., 2022), and ST-Transformer (Song et al., 2021a), implemented following the BIOT repository (Yang et al., 2023) unless official results are available. For foundation-model baselines, we evaluate BIOT (Yang et al., 2023) and LaBraM (Jiang

Table 1: Results on TUEV with different models and model complexity.

| Methods | Model Size (#Params) | Time / Epoch (m) | Balanced Acc. | Cohen's Kappa | Weighted F1 |
|---|---|---|---|---|---|
| SPaRCNet | 0.79M | 0.06 | $0.4161 \pm 0.0262$ | $0.4233 \pm 0.0181$ | $0.7024 \pm 0.0104$ |
| ContraWR | 1.6M | 0.07 | $0.4384 \pm 0.0349$ | $0.3912 \pm 0.0237$ | $0.6893 \pm 0.0136$ |
| CNN-Transformer | 3.2M | 0.12 | $0.4087 \pm 0.0161$ | $0.3815 \pm 0.0134$ | $0.6854 \pm 0.0293$ |
| FFCL | 2.4M | 0.12 | $0.3979 \pm 0.0104$ | $0.3732 \pm 0.0188$ | $0.6783 \pm 0.0120$ |
| ST-Transformer | 3.5M | 0.03 | $0.3984 \pm 0.0228$ | $0.3765 \pm 0.0306$ | $0.6823 \pm 0.0190$ |
| *BIOT (non-pretrained)* | | | | | |
| BIOT | 3.2M | 0.12 | $0.4682 \pm 0.0125$ | $0.4482 \pm 0.0285$ | $0.7085 \pm 0.0184$ |
| BIOT+iSQRT-COV | $3.2M+\Delta 33.1K$ | 0.81 | $0.4480 \pm 0.0131$ | $0.3544 \pm 0.0221$ | $0.6323 \pm 0.0203$ |
| BIOT+SVD-Padé | $3.2M+\Delta 33.1K$ | 10.61 | $0.4347 \pm 0.0133$ | $0.4091 \pm 0.0293$ | $0.6906 \pm 0.0137$ |
| BIOT+iSICE | $3.2M+\Delta 33.1K$ | 4.71 | $0.4630 \pm 0.0143$ | $0.4563 \pm 0.0197$ | $0.7140 \pm 0.0167$ |
| BIOT+RHOP | $3.2M+\Delta 1.3K$ | 0.53 | $\mathbf{0.5355 \pm 0.0189}$ | $\mathbf{0.5177 \pm 0.0252}$ | $\mathbf{0.7466 \pm 0.0084}$ |
| *BIOT (pretrained)* | | | | | |
| BIOT | 3.2M | 0.12 | $0.5281 \pm 0.0225$ | $0.5273 \pm 0.0249$ | $0.7492 \pm 0.0082$ |
| BIOT+iSQRT-COV | $3.2M+\Delta 33.1K$ | 0.81 | $0.4683 \pm 0.0146$ | $0.3563 \pm 0.0218$ | $0.6530 \pm 0.0187$ |
| BIOT+SVD-Padé | $3.2M+\Delta 33.1K$ | 10.62 | $0.4372 \pm 0.0176$ | $0.4826 \pm 0.0162$ | $0.7399 \pm 0.0171$ |
| BIOT+iSICE | $3.2M+\Delta 33.1K$ | 4.73 | $0.5358 \pm 0.0241$ | $0.5245 \pm 0.0203$ | $0.7534 \pm 0.0094$ |
| BIOT+RHOP | $3.2M+\Delta 1.3K$ | 0.73 | $\mathbf{0.5572 \pm 0.0201}$ | $\mathbf{0.5460 \pm 0.0212}$ | $\mathbf{0.7565 \pm 0.0074}$ |
| *LaBraM-Base (pretrained)* | | | | | |
| LaBraM-Base | 5.8M | 1.03 | $\mathbf{0.6409 \pm 0.0065}$ | $0.6637 \pm 0.0093$ | $0.8312 \pm 0.0052$ |
| LaBraM-Base+iSQRT-COV | $5.8M+\Delta 20.3K$ | 3.23 | $0.6236 \pm 0.0226$ | $0.6147 \pm 0.0234$ | $0.8062 \pm 0.0094$ |
| LaBraM-Base+SVD-Padé | $5.8M+\Delta 20.3K$ | 6.93 | $0.5605 \pm 0.0217$ | $0.5798 \pm 0.0289$ | $0.7900 \pm 0.0101$ |
| LaBraM-Base+iSICE | $5.8M+\Delta 20.3K$ | 13.67 | $0.6405 \pm 0.0239$ | $0.6134 \pm 0.0302$ | $0.8182 \pm 0.0091$ |
| LaBraM-Base+RHOP | $5.8M+\Delta 0.9K$ | 2.25 | $0.6380 \pm 0.0056$ | $\mathbf{0.6785 \pm 0.0079}$ | $\mathbf{0.8420 \pm 0.0038}$ |

Table 2: Results on TUAB with model complexity.

| Methods | Model Size (#Params) | Time / Epoch (m) | Balanced Acc. | AUC-PR | AUROC |
|---|---|---|---|---|---|
| SPaRCNet | 0.79M | 0.43 | $0.7896 \pm 0.0018$ | $0.8414 \pm 0.0018$ | $0.8676 \pm 0.0012$ |
| ContraWR | 1.6M | 0.48 | $0.7746 \pm 0.0041$ | $0.8421 \pm 0.0104$ | $0.8456 \pm 0.0074$ |
| CNN-Transformer | 3.2M | 0.76 | $0.7777 \pm 0.0022$ | $0.8433 \pm 0.0039$ | $0.8461 \pm 0.0013$ |
| FFCL | 2.4M | 0.86 | $0.7848 \pm 0.0038$ | $0.8448 \pm 0.0065$ | $0.8569 \pm 0.0051$ |
| ST-Transformer | 3.5M | 0.24 | $0.7966 \pm 0.0023$ | $0.8521 \pm 0.0026$ | $0.8707 \pm 0.0019$ |
| *BIOT (non-pretrained)* | | | | | |
| BIOT | 3.2M | 1.36 | $0.7925 \pm 0.0035$ | $0.8707 \pm 0.0087$ | $0.8691 \pm 0.0033$ |
| BIOT+iSQRT-COV | $3.2M+\Delta 33.1K$ | 4.78 | $0.7983 \pm 0.0045$ | $0.8684 \pm 0.0051$ | $0.8659 \pm 0.0055$ |
| BIOT+SVD-Padé | $3.2M+\Delta 33.1K$ | 50.71 | $0.7503 \pm 0.0051$ | $0.8274 \pm 0.0058$ | $0.8270 \pm 0.0068$ |
| BIOT+iSICE | $3.2M+\Delta 33.1K$ | 10.47 | $0.7959 \pm 0.0057$ | $0.8792 \pm 0.0023$ | $0.8815 \pm 0.0043$ |
| BIOT+RHOP | $3.2M+\Delta 1.0K$ | 3.64 | $\mathbf{0.7993 \pm 0.0031}$ | $\mathbf{0.8719 \pm 0.0084}$ | $\mathbf{0.8765 \pm 0.0031}$ |
| *BIOT (pretrained)* | | | | | |
| BIOT | 3.2M | 1.09 | $0.7959 \pm 0.0057$ | $0.8792 \pm 0.0023$ | $0.8815 \pm 0.0043$ |
| BIOT+iSQRT-COV | $3.2M+\Delta 33.1K$ | 4.78 | $0.7819 \pm 0.0044$ | $0.8590 \pm 0.0028$ | $0.8598 \pm 0.0039$ |
| BIOT+SVD-Padé | $3.2M+\Delta 33.1K$ | 54.38 | $0.7532 \pm 0.0064$ | $0.8274 \pm 0.0043$ | $0.8270 \pm 0.0056$ |
| BIOT+iSICE | $3.2M+\Delta 33.1K$ | 10.50 | $0.7976 \pm 0.0097$ | $0.8617 \pm 0.0046$ | $0.8739 \pm 0.0040$ |
| BIOT+RHOP | $3.2M+\Delta 1.3K$ | 3.69 | $\mathbf{0.8102 \pm 0.0027}$ | $\mathbf{0.8833 \pm 0.0079}$ | $\mathbf{0.8864 \pm 0.0033}$ |
| *LaBraM-Base (pretrained)* | | | | | |
| LaBraM-Base | 5.8M | 11.54 | $0.8140 \pm 0.0019$ | $0.8965 \pm 0.0016$ | $0.9022 \pm 0.0009$ |
| LaBraM-Base+iSQRT-COV | $5.8M+\Delta 20.3K$ | 19.07 | $0.8188 \pm 0.0023$ | $0.9039 \pm 0.0018$ | $0.9060 \pm 0.0012$ |
| LaBraM-Base+SVD-Padé | $5.8M+\Delta 20.3K$ | 31.23 | $0.8202 \pm 0.0017$ | $0.9062 \pm 0.0014$ | $0.9072 \pm 0.0011$ |
| LaBraM-Base+iSICE | $5.8M+\Delta 20.3K$ | 67.77 | $0.8183 \pm 0.0018$ | $0.9037 \pm 0.0016$ | $0.9059 \pm 0.0016$ |
| LaBraM-Base+RHOP | $5.8M+\Delta 4.6K$ | 21.48 | $\mathbf{0.8244 \pm 0.0012}$ | $\mathbf{0.9078 \pm 0.0012}$ | $\mathbf{0.9105 \pm 0.0011}$ |

et al., 2024), using their released code and checkpoints. Since only LaBraM-Base provides public weights, we fine-tune this variant in all experiments, while BIOT is tested both with and without pre-training. In addition, we compare RHOP against several representative global covariance pooling (GCP) heads, including iSICE (Rahman et al., 2023), iSQRT-COV (Li et al., 2018), and SVD-Padé (Song et al., 2021b). For fairness, given EEG inputs, all backbones produce spatiotemporal tokens and a CLS token. GCP variants then compute covariance descriptors, which we vectorize via the upper triangular (including the diagonal) before classification with a linear layer.

**Metrics.** We report results using five widely adopted evaluation metrics: (1) **Balanced Accuracy**, defined as the mean recall across classes, applied to both binary and multi-class settings. (2) **AUC-PR**, the area under the precision–recall curve, for binary classification. (3) **AUROC**, the area under the receiver operating characteristic curve, also for binary classification. (4) **Cohen's Kappa**, which measures agreement beyond chance by comparing observed and expected accuracies in a contingency table, is used in multi-class tasks. (5) **Weighted F1**, the weighted harmonic mean of precision and recall, used for multi-class evaluation. For monitoring, AUROC is adopted in binary classification experiments, while Cohen's Kappa is used in multi-class settings.

Table 3: Results on BCIC2B with different models and model complexity.

| Methods | Model Size (#Params) | Time / Epoch (m) | Balanced Acc. | AUC-PR | AUROC |
|---|---|---|---|---|---|
| LaBraM-Base | 5.8M | 0.04 | $0.6840 \pm 0.0059$ | $0.7405 \pm 0.0087$ | $0.7472 \pm 0.0054$ |
| LaBraM-Base+iSQRT-COV | 5.8M+$\Delta$20.3K | 0.38 | $0.6642 \pm 0.0061$ | $0.7210 \pm 0.0079$ | $0.7285 \pm 0.0058$ |
| LaBraM-Base+SVD-Padé | 5.8M+$\Delta$20.3K | 0.15 | $0.6629 \pm 0.0056$ | $0.7198 \pm 0.0083$ | $0.7267 \pm 0.0061$ |
| LaBraM-Base+iSICE | 5.8M+$\Delta$20.3K | 0.97 | $0.6871 \pm 0.0058$ | $0.7433 \pm 0.0082$ | $0.7510 \pm 0.0055$ |
| LaBraM-Base+RHOP | 5.8M+$\Delta$0.5K | 0.05 | $\mathbf{0.6901 \pm 0.0057}$ | $\mathbf{0.7485 \pm 0.0079}$ | $\mathbf{0.7587 \pm 0.0059}$ |

Table 4: Results on PhysioP300 with model complexity.

| Methods | Model Size (#Params) | Time / Epoch (m) | Balanced Acc. | AUC-PR | AUROC |
|---|---|---|---|---|---|
| LaBraM-Base | 5.8M | 0.05 | $0.6327 \pm 0.019$ | $0.6565 \pm 0.024$ | $0.6893 \pm 0.027$ |
| LaBraM-Base+iSQRT-COV | 5.8M+$\Delta$20.3K | 0.08 | $0.6170 \pm 0.016$ | $0.6560 \pm 0.019$ | $0.6813 \pm 0.012$ |
| LaBraM-Base+SVD-Padé | 5.8M+$\Delta$20.3K | 0.16 | $0.6323 \pm 0.015$ | $0.6400 \pm 0.020$ | $0.6751 \pm 0.014$ |
| LaBraM-Base+iSICE | 5.8M+$\Delta$20.3K | 0.41 | $0.6138 \pm 0.018$ | $0.6350 \pm 0.022$ | $0.6427 \pm 0.017$ |
| LaBraM-Base+RHOP | 5.8M+$\Delta$0.4K | 0.08 | $\mathbf{0.6517 \pm 0.018}$ | $\mathbf{0.6630 \pm 0.016}$ | $\mathbf{0.7044 \pm 0.024}$ |

## 5.2 EVALUATIONS

**Training from scratch.**  On TUEV, as reported in Tab. 1, RHOP yields larger gains: balanced accuracy increases from 46.82% to 53.55%, Cohen's Kappa from 44.82% to 51.77%, and Weighted F1 from 70.85% to 74.66%, with only 0.53 minutes per epoch. On TUAB, as shown in Tab. 2, BIOT with RHOP improves balanced accuracy from 79.25% to 79.93% and AUROC from 86.91% to 87.65%, while reducing per-epoch time to 3.64 minutes compared with 10.47 minutes for iSICE and 50.71 minutes for SVD-Padé. These results highlight that RHOP is particularly effective without pretraining, as it preserves spatiotemporal structure and high-order statistics.

**Full fine-tuning.**  On TUEV, as shown in Tab. 1, RHOP reaches a higher Cohen's Kappa of 67.85% and a Weighted F1 of 84.20%, while maintaining strong balanced accuracy. On TUAB, as reported in Tab. 2, LaBraM-Base with RHOP improves balanced accuracy from 81.40% to 82.44% and pushes AUROC to 91.05%. These improvements confirm that even strong pretrained backbones benefit from RHOP, which captures dependencies overlooked by conventional heads.

**Linear-head tuning.**  With frozen backbones, RHOP still surpasses all baselines. On BCIC2B, LaBraM-Base with RHOP reaches 69.01% balanced accuracy and 75.87% AUROC, outperforming all GCP heads, as given in Tab. 3. On PhysioP300, it improves balanced accuracy from 63.27% to 65.17% and AUROC from 68.93% to 70.44%, as presented in Tab. 4. These results demonstrate RHOP's ability to deliver discriminative representations even when backbone weights are frozen, which makes it suitable for rapid adaptation with minimal compute.

**Why RHOP outperforms GCP?**  Classical GCP methods such as iSQRT-COV, iSICE, and SVD-Padé collapse all tokens into a single covariance matrix, which discards temporal and channel hierarchy. RHOP instead normalizes tokens' covariances to correlation form, embeds both means and normalized covariances on the SPD manifold, and aggregates via a Riemannian Gaussian defined by a Fréchet mean and tangent-space covariance. This preserves scale-invariant dependencies, aligns computation with manifold geometry, and emphasizes direct relationships through a sparse inverse-covariance layer. The result is a more faithful global representation that consistently outperforms GCP, especially in challenging setups such as TUEV with training from scratch.

**Efficiency.**  RHOP delivers accuracy gains with minimal overhead. On TUAB, LaBraM-Base with RHOP trains in 21.48 min per epoch with only 4.6K additional parameters, compared with 67.77 min for iSICE and 31.23 minutes for SVD-Padé as reported in Tab. 2. BIOT with RHOP requires just 3.69 min per epoch when pretrained, and only 0.53 min per epoch when trained from scratch on TUEV, far below iSICE and SVD-Padé, as shown in Tab. 1. On BCIC2B and PhysioP300, RHOP adds less than 1K parameters and only 0.01 min per epoch, while still improving all metrics, as shown in Tabs. 3 and 4. Overall, RHOP combines strong accuracy, scale robustness, and geometric fidelity with thousand-level parameter overhead and negligible training cost, making it a practical plug-and-play head for EEG foundation backbones.

## 5.3 ABLATION STUDY

**Quotient Gaussian Embedding vs Gaussian Embedding.**  Fig. 2 compares quotient Gaussian embedding with conventional Gaussian embedding. The only difference is that conventional em-

Table 5: Ablation study on different components, where QGE denotes Quotient Gaussian Embedding, RGE denotes Riemannian Gaussian Embedding, and CLS indicates whether concatenation with the classification head is applied.

| QGE | RGE | SICE | CLS | TUAB | | | TUEV | | |
|---|---|---|---|---|---|---|---|---|---|
| | | | | Balanced Acc. | AUC-PR | AUROC | Balanced Acc. | Cohen's Kappa | Weighted F1 |
| ✗ | ✗ | ✗ | ✗ | $0.8140 \pm 0.0019$ | $0.8965 \pm 0.0016$ | $0.9022 \pm 0.0009$ | $\mathbf{0.6409 \pm 0.0065}$ | $0.6637 \pm 0.0093$ | $0.8312 \pm 0.0052$ |
| ✓ | ✗ | ✗ | ✗ | $0.8175 \pm 0.0018$ | $0.9002 \pm 0.0015$ | $0.9048 \pm 0.0010$ | $0.6325 \pm 0.0062$ | $0.6669 \pm 0.0090$ | $0.8331 \pm 0.0051$ |
| ✓ | ✓ | ✗ | ✗ | $0.8209 \pm 0.0016$ | $0.9031 \pm 0.0014$ | $0.9069 \pm 0.0010$ | $0.6355 \pm 0.0060$ | $0.6712 \pm 0.0087$ | $0.8365 \pm 0.0049$ |
| ✓ | ✓ | ✓ | ✗ | $0.8227 \pm 0.0014$ | $0.9056 \pm 0.0013$ | $0.9088 \pm 0.0010$ | $0.6341 \pm 0.0058$ | $0.6749 \pm 0.0084$ | $0.8391 \pm 0.0047$ |
| ✓ | ✓ | ✓ | ✓ | $\mathbf{0.8244 \pm 0.0012}$ | $\mathbf{0.9078 \pm 0.0012}$ | $\mathbf{0.9105 \pm 0.0011}$ | $0.6380 \pm 0.0056$ | $\mathbf{0.6785 \pm 0.0079}$ | $\mathbf{0.8420 \pm 0.0038}$ |

Table 6: Ablation of $(k, k')$ on LaBraM-Base.

| $k$ | $k'$ | TUAB | | | TUEV | | |
|---|---|---|---|---|---|---|---|
| | | Balanced Acc. | AUC-PR | AUROC | Balanced Acc. | Cohen's Kappa | Weighted F1 |
| 1 | 1 | $0.8065 \pm 0.0019$ | $0.8973 \pm 0.0017$ | $0.8988 \pm 0.0012$ | $0.6457 \pm 0.0121$ | $0.5772 \pm 0.0203$ | $0.7876 \pm 0.0114$ |
| | 2 | $0.8193 \pm 0.0021$ | $0.9054 \pm 0.0016$ | $0.9064 \pm 0.0013$ | $0.6339 \pm 0.0184$ | $0.6078 \pm 0.0227$ | $0.8062 \pm 0.0102$ |
| | 3 | $0.8199 \pm 0.0020$ | $0.9054 \pm 0.0015$ | $0.9060 \pm 0.0012$ | $\mathbf{0.6726 \pm 0.0137}$ | $0.6456 \pm 0.0181$ | $0.8224 \pm 0.0097$ |
| 2 | 1 | $0.8171 \pm 0.0018$ | $0.8975 \pm 0.0018$ | $0.9013 \pm 0.0012$ | $0.6631 \pm 0.0116$ | $0.6471 \pm 0.0193$ | $0.8252 \pm 0.0089$ |
| | 2 | $\mathbf{0.8249 \pm 0.0022}$ | $0.9044 \pm 0.0014$ | $0.9071 \pm 0.0011$ | $0.6459 \pm 0.0128$ | $0.6008 \pm 0.0251$ | $0.7970 \pm 0.0095$ |
| | 3 | $0.8163 \pm 0.0017$ | $0.9058 \pm 0.0016$ | $0.9057 \pm 0.0011$ | $0.6440 \pm 0.0137$ | $0.6196 \pm 0.0228$ | $0.8145 \pm 0.0117$ |
| 3 | 1 | $0.8183 \pm 0.0019$ | $0.9035 \pm 0.0015$ | $0.9062 \pm 0.0011$ | $0.6659 \pm 0.0123$ | $0.6317 \pm 0.0273$ | $0.8176 \pm 0.0135$ |
| | 2 | $0.8193 \pm 0.0020$ | $0.9008 \pm 0.0016$ | $0.9039 \pm 0.0012$ | $0.6372 \pm 0.0161$ | $0.6220 \pm 0.0234$ | $0.8135 \pm 0.0092$ |
| | 3 | $0.8244 \pm 0.0012$ | $\mathbf{0.9078 \pm 0.0012}$ | $\mathbf{0.9105 \pm 0.0011}$ | $0.6380 \pm 0.0056$ | $\mathbf{0.6785 \pm 0.0079}$ | $\mathbf{0.8420 \pm 0.0038}$ |

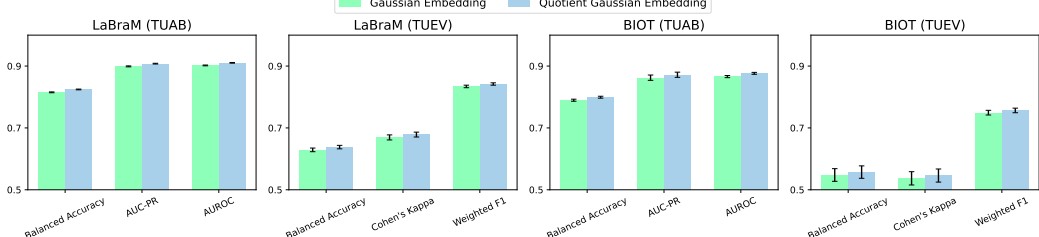

Figure 2: Quotient Gaussian embedding vs Gaussian embedding on TUEV and TUAB.

bedding (Nguyen, 2021) directly uses the covariance matrix without normalization. On both TUAB and TUEV, the quotient form consistently outperforms the raw covariance representation, confirming that scale normalization is essential for robust EEG descriptors.

**Component analysis.** Tab. 5 evaluates the contribution of each RHOP component. QGE alone improves performance, RGE further enhances high-order modeling, and SICE highlights partial correlations. The best results are obtained when CLS fusion is added, which combines semantic and statistical cues. Overall, each component provides complementary gains, and the full RHOP yields the strongest improvements.

**Embedding dimensions.** The parameters $k$ and $k'$ control the augmentation of mean vectors and the embedding of Riemannian statistics in Eqs. (9) and (11). Tab. 6 reports their influence on TUAB and TUEV. Non-trivial choices with $k, k' > 0$ consistently outperform the baseline, and the optimal configuration varies across datasets. TUAB achieves the highest AUROC with $k = 3, k' = 3$, while TUEV obtains the best Cohen's Kappa with $k = 3, k' = 3$ and the best balanced accuracy with $k = 1, k' = 3$. Increasing $k$ or $k'$ expands the SPD embedding dimension and amplifies the mean-related terms, which reduces the relative influence of the covariance structure and introduces additional computational and numerical burdens. As a result, moderate settings such as $(k = 3, k' = 3)$ achieve the best trade-off between representational expressiveness and model stability.

**Covariance normalization.** Tab. 7 examines why iSICE is selected as the final covariance normalization method. Replacing iSICE with iSQRT-COV or SVD-Padé reduces performance, while iSICE achieves the best results on TUAB and TUEV. The reason is that iSICE imposes sparsity during normalization, which regularizes high-dimensional features. This property is particularly ef-

Table 7: Comparison of different GCP heads on LaBraM-Base.

| Normalization | TUAB | | | TUEV | | |
|---|---|---|---|---|---|---|
| | Balanced Acc. | AUC-PR | AUROC | Balanced Acc. | Cohen's Kappa | Weighted F1 |
| SVD-Padé | $0.8190 \pm 0.0016$ | $0.9012 \pm 0.0014$ | $0.9058 \pm 0.0010$ | $0.6368 \pm 0.0071$ | $0.6710 \pm 0.0085$ | $0.8365 \pm 0.0049$ |
| iSQRT-COV | $0.8181 \pm 0.0017$ | $0.9001 \pm 0.0015$ | $0.9051 \pm 0.0011$ | $0.6375 \pm 0.0068$ | $0.6692 \pm 0.0091$ | $0.8350 \pm 0.0047$ |
| iSICE | $\mathbf{0.8244 \pm 0.0012}$ | $\mathbf{0.9078 \pm 0.0012}$ | $\mathbf{0.9105 \pm 0.0011}$ | $\mathbf{0.6380 \pm 0.0056}$ | $\mathbf{0.6785 \pm 0.0079}$ | $\mathbf{0.8420 \pm 0.0038}$ |

Table 8: Gaussianity statistics of token features after random projection. SW-$p$: Shapiro–Wilk $p$-value; Prop.: fraction of projections with $p > 0.05$ (non-rejection of Gaussianity).

(a) RHOP temporal-segment features.

| Model | Dataset | SW-$p$ | Prop.$(p > 0.05)$ | Skew | Kurt |
|---|---|---|---|---|---|
| LaBraM | TUAB | 0.49 | 0.94 | $-0.13$ | 0.19 |
| | TUEV | 0.47 | 0.89 | $-0.10$ | 0.15 |
| BIOT | TUAB | 0.46 | 0.91 | $-0.07$ | 0.10 |
| | TUEV | 0.44 | 0.85 | $-0.09$ | 0.12 |

(b) GCP token-level aggregated features.

| Model | Dataset | SW-$p$ | Prop.$(p > 0.05)$ | Skew | Kurt |
|---|---|---|---|---|---|
| LaBraM | TUAB | 0.01 | 0.05 | 0.14 | 18.94 |
| | TUEV | 0.01 | 0.03 | 0.25 | 19.54 |
| BIOT | TUAB | 0.02 | 0.03 | 0.22 | 19.16 |
| | TUEV | 0.01 | 0.00 | 0.14 | 19.97 |

fective for EEG decoding, where signals are noisy and correlations are often spurious, making iSICE a more robust choice than dense normalization approaches.

## 5.4 GAUSSIANITY ANALYSIS

In this section, we evaluate the Gaussianity of token features extracted from LaBraM and BIOT on TUAB and TUEV, aiming to understand the statistical behavior of foundation model activations.

**Testing procedure.** For each dataset, we randomly sample $10\,000$ data samples, extract their token representations, and assess the distributional properties of the temporal feature vectors. Each token feature is projected onto multiple random one-dimensional directions, a standard approach for testing multivariate normality. For each projection, we compute the Shapiro–Wilk $p$-value, skewness, and excess kurtosis, and then average these quantities.

**Discussion.** As summarized in Tab. 8(a), we evaluate Gaussianity along the temporal dimension where RHOP computes covariance. The resulting temporal-segment features exhibit near-Gaussian behavior: Shapiro–Wilk $p$-values are moderate, $85\%$–$94\%$ of projections satisfy $p > 0.05$, skewness is close to zero, and excess kurtosis remains small. These results indicate that RHOP's temporal-segment representations follow a Gaussian suitable for second-order modeling. In contrast, the GCP features in Tab. 8(b) exhibit strong non-Gaussian behavior. GCP computes covariance after flattening and aggregating spatial and temporal tokens, thereby mixing heterogeneous EEG components within a single representation. When Gaussianity is assessed along this mixed token axis, the statistics deviate markedly from normality: the Shapiro–Wilk $p$-values are near zero, fewer than $5\%$ of projections satisfy $p > 0.05$, and the excess kurtosis reaches extremely high values. These results indicate heavy-tailed, asymmetric, and multimodal structure, reflecting the statistical complexity introduced by combining spatial and temporal information in a single representation. By contrast, RHOP evaluates covariance within coherent temporal segments rather than across mixed tokens. It operates on statistically well-behaved representations, leading to more stable covariance estimates and improved downstream performance.

## 6 CONCLUSION

This work introduced Riemannian High-Order Pooling (RHOP), a geometry-aware classification head designed to complement large EEG foundation backbones. By embedding per-token quotient Gaussians on the SPD manifold and aggregating them into a Riemannian Gaussian descriptor, RHOP preserves scale-invariant dependencies and captures high-order spatiotemporal interactions. Extensive experiments across abnormal detection, epileptic event classification, motor imagery, and event-related potentials demonstrated that RHOP consistently improves accuracy, robustness, and efficiency over global average pooling and classical covariance pooling. These gains persist across different training regimes, including training from scratch, full fine-tuning, and linear probing, confirming that RHOP provides a principled and effective bridge between Riemannian statistics and foundation models. More broadly, this study highlights the value of geometric inductive bias in large-scale EEG modeling and supports geometry-aware analysis of brain signals.

ETHICS STATEMENT

This work adheres to the ICLR Code of Ethics. All experiments were conducted on publicly available datasets with appropriate licenses, and no personally identifiable or sensitive information was involved.

REPRODUCIBILITY STATEMENT

We have made every effort to ensure reproducibility. The datasets used are publicly available. Model architectures, training procedures, and evaluation metrics are detailed in Sections 4 and 5, with additional implementation details and hyperparameters provided in the appendix. Upon acceptance, we will release the complete source code and instructions for reproducing all experiments.

ACKNOWLEDGE

This work was supported in part by the National Natural Science Foundation of China (62306127, 62332008), the Natural Science Foundation of Jiangsu Province (BK20231040), the Fundamental Research Funds for the Central Universities (JUSRP124015), the Postgraduate Research & Practice Innovation Program of Jiangsu Province (SJCX25_1319), the EU Horizon project ELLIOT (101214398), the FIS project GUIDANCE (FIS2023-03251), the National Key R&D Program of China (2023YFF1105102, 2023YFF1105105), the Zhejiang Leading Innovative and Entrepreneur Team Introduction Program (2024R01007), and the "Pioneer" and "Leading Goose" Research and Development Program of Zhejiang (2025C02077).

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

## A  THE USE OF LARGE LANGUAGE MODELS (LLMS)

We used large language models (LLMs) only for limited assistance in language polishing and for implementing minor parts of the code. All research ideas, experiment design, analysis, and conclusions were conceived and validated by the authors, who remain fully responsible for the content of this work.

## B  NOTATIONS

For better clarity, we summarize all the notations used in this paper in Tab. 9.

Table 9: Notations used in this paper.

| Symbol | Meaning |
|---|---|
| $\mathcal{S}_n^+$ | Set of $n \times n$ symmetric positive definite matrices |
| $d_{\mathrm{AIM}}(P, Q)$ | Affine invariant Riemannian distance on $\mathcal{S}_n^+$ |
| $\mathrm{Exp}_P(S)$ | Exponential map at $P \in \mathcal{S}_n^+$ for tangent vector $S \in T_P\mathcal{S}_n^+$ |
| $\mathrm{Log}_P(Q)$ | Logarithm map at $P \in \mathcal{S}_n^+$ for $Q \in \mathcal{S}_n^+$ |
| $T_P\mathcal{S}_n^+$ | Tangent space of $\mathcal{S}_n^+$ at $P$ |
| $\|\cdot\|_F$ | Frobenius norm |
| $\mathrm{WFM}(\{w_i\}, \{P_i\})$ | Weighted Fréchet mean on $\mathcal{S}_n^+$ with weights $\{w_i\}$ |
| $w_i$ | Nonnegative weights with $\sum_i w_i = 1$ |
| $\mathcal{N}(n)$ | Family of $n$-dimensional Gaussians parameterized by $(\Sigma, \mu)$ |
| $\Sigma, \mu$ | Covariance matrix and mean vector of a Gaussian |
| $k, k'$ | Hyperparameters for quotient and Riemannian embedding dimensions |

## C  DATASET DESCRIPTION AND PREPROCESSING

In this section, we provide additional implementation details to ensure full reproducibility of our experiments, including dataset descriptions, preprocessing steps, and training configurations that were not covered in the main paper.

### C.1  THE TUAB DATASET

**Description.**  The TUAB dataset[1] is a large clinical EEG corpus in which each recording is labeled as either normal or abnormal. The raw EEGs are 23-channel clinical recordings sampled at 256 Hz. After segmentation into non-overlapping 10-second windows, the corpus provides 409,455 samples for binary normal/abnormal classification.

**Preprocessing.**  For experiments using the BIOT model, we adopt the preprocessing pipeline provided in BIOT (Yang et al., 2023). All recordings are processed using 16 bipolar montage channels defined under the 10–20 system. Each EEG signal is first resampled to 200 Hz and then transformed into a standardized set of 16 bipolar derivations (Yang et al., 2023), which specify fixed electrode pairs (e.g., FP1–F7, F7–T3, T3–T5, etc.). Segments shorter than 10 seconds are discarded. For experiments using LaBraM, we adopt the preprocessing protocol defined in (Jiang et al., 2024). The EEG signals are first band-pass filtered between 0.1 Hz and 75 Hz to remove low-frequency drift, followed by a 50 Hz notch filter to suppress power-line interference. All recordings are then resampled to 200 Hz. Since raw EEG values typically lie within $[-0.1, 0.1]$ mV, we normalize the amplitude by scaling the unit to 0.1 mV such that the resulting signals fall approximately within $[-1, 1]$. Each valid 10-second window inherits the normal/abnormal label of its source recording and is stored as an individual sample for downstream training, validation, and evaluation.

**Experimental Configuration.**  We adopt a subject-independent evaluation protocol. Since the official subject-wise split already provides separate training and test partitions, we further divide the training portion into an 80%–20% train–validation split, following prior work (Yang et al., 2023; Jiang et al., 2024). The model achieving the highest validation performance is selected for final testing. Each method is evaluated using its own preprocessing pipeline and corresponding channel configuration: 16 channels for BIOT and 23 channels for LaBraM. To improve statistical reliability, all experiments are repeated three times with different random seeds, and we report the mean and standard deviation across runs.

---

[1] https://isip.piconepress.com/projects/tuh_eeg/html/downloads.shtml

### C.2 THE TUEV DATASET

**Description.** The TUEV[2] corpus is a clinically annotated subset of TUEG that categorizes EEG segments into six event types: spike and sharp wave (SPSW), generalized periodic epileptiform discharges (GPED), periodic lateralized epileptiform discharges (PLED), eye movement (EYEM), artifact (ARTF), and background (BCKG). All recordings contain 23 EEG channels sampled at 256 Hz, and the dataset provides 112,491 non-overlapping 5-second segments for multiclass classification.

**Preprocessing.** We adopt the same preprocessing pipeline as in TUAB. Each EEG recording is resampled to 200 Hz and transformed into the standardized 16-channel bipolar montage used in BIOT Yang et al. (2023).Segments shorter than 5 s are discarded. Each 5-second segment inherits the event label of its source annotation and is stored as an individual sample for downstream training, validation, and evaluation.

**Experimental Configuration.** Following the official subject-wise split, the training portion is further divided into an 80%–20% train–validation split, and the model with the best validation accuracy is used for final testing. All experiments are repeated three times with different random seeds, and the mean and standard deviation are reported.

### C.3 THE BCIC2B DATASET

**Description.** The BCIC2B motor-imagery dataset[3] contains recordings from 9 participants and includes both EEG and EOG modalities. EEG signals were collected from three bipolar channels (C3, Cz, C4) at a sampling rate of 250 Hz, while ocular activity was monitored using three monopolar EOG electrodes with a dynamic range of $\pm 1$ mV. Each participant completed five experimental runs: the first two were screening sessions without feedback, and the remaining three provided feedback. The motor imagery task involved imagining either left-hand or right-hand movements (class 1 and class 2). For the screening phase, each subject was recorded on two separate days within a two-week interval, and every session consisted of 120 trials evenly distributed across the two classes.

**Preprocessing.** We discard all ocular channels (`EOG:ch01`, `EOG:ch02`, `EOG:ch03`) prior to further processing. The EEG signals are band-pass filtered within 0–38 Hz, resampled to 200 Hz, and subsequently normalized using EA normalization He & Wu (2019), which is widely adopted for motor-imagery datasets.

**Experimental Configuration.** We employ a subject-independent evaluation. Since the BCIC2B dataset contains recordings from 9 subjects, the experiments follow a 9-fold LOSO cross-validation protocol. In each fold, one subject (e.g., `sub1`) is designated as the test set, and the remaining subjects (`sub2--sub9`) are used for training. To enhance statistical reliability, every experiment is repeated 3 times with different random seeds, and the mean and standard deviation are reported.

### C.4 THE PHYSIOP300 DATASET

**Description.** The PhysioNetP300 dataset[4] includes recordings from 12 participants performing a P300-based character-spelling task using a Donchin speller. EEG was acquired with a 64-channel BioSemi ActiveTwo system at a sampling rate of 2048 Hz. Each subject was asked to spell 20 characters. For each run, a target character was randomly chosen, and the rows and columns of a standard 6×6 matrix were intensified for 100 ms with 50 ms intervals (SOA of 150 ms), resulting in roughly 20 flashes of the target per run. During the sequence, subjects focused on the designated character and counted its occurrences.

**Preprocessing.** We retain all 64 EEG channels from the BioSemi montage and first convert the raw signals into uniform units. The data are band-pass filtered within 0–120 Hz and resampled to 200 Hz. For each trial, we use a 2-second window beginning 0.7 s before the flash onset, which corresponds approximately to a $[-0.1, 2]$ s peri-stimulus interval after resampling. Each extracted epoch is assigned a binary label (target or non-target) depending on whether the intensified row or column contains the designated character.

---

[2] https://isip.piconepress.com/projects/tuh_eeg/html/downloads.shtml
[3] https://www.bbci.de/competition/iv/#datasets
[4] https://physionet.org/content/erpbci/1.0.0/

Table 10: Dataset–backbone specific hyperparameters for downstream training.

| Hyperparameters | TUAB/TUEV | | BCIC2B | | PhysioNetP300 | |
|---|---|---|---|---|---|---|
| | LaBraM | BIOT | LaBraM | BIOT | LaBraM | BIOT |
| Batch size | 64 | 64 | 64 | 64 | 64 | 64 |
| LR scheduler | Cosine | Cosine | OneCycle | OneCycle | OneCycle | OneCycle |
| Start learning rate | — | — | 1.5e-5 | 1.5e-5 | 3.0e-5 | 3.0e-5 |
| Peak/Max learning rate | 5e-4 | 2.5e-3 | 4.0e-4 | 4.0e-4 | 8.0e-4 | 8.0e-4 |
| Minimal learning rate | 1e-6 | 1e-6 | 1.5e-7 | 1.5e-7 | 3.0e-7 | 3.0e-7 |
| Optimizer | AdamW | Adam | AdamW | AdamW | AdamW | AdamW |
| Adam $\beta$ | 0.9, 0.999 | 0.9, 0.999 | 0.9, 0.999 | 0.9, 0.999 | 0.9, 0.999 | 0.9, 0.999 |
| Weight decay | 0.05 | 0.0005 | 0.05 | 0.05 | 0.05 | 0.05 |
| Total epochs | 50 | 50 | 100 | 100 | 100 | 100 |
| Warmup epochs | 5 | 0 | — | — | — | — |
| Dropout | 0 | 0.2 | 0 | 0.2 | 0 | 0.2 |
| Drop path | 0.1 | 0 | 0 | 0 | 0 | 0 |
| Layer-wise LR decay | 0.65 | — | - | — | - | — |
| Label smoothing (multi-class) | 0.1 | — | 0.1 | — | 0.1 | — |
| Sliding-window step | — | — | — | — | 125 ms | 125 ms |

**Experimental Configuration.** We follow the configuration used in BENDR Kostas et al. (2021) and exclude subjects 8, 10, and 12, leaving data from the remaining 9 participants. A subject-independent evaluation is employed, using a 9-fold LOSO cross-validation protocol in which one subject is held out for testing while the others are used for training. During fine-tuning, the PhysioNetP300 data are processed using a 125 ms sliding window to extract temporally localized ERP segments for classification.

# D HYPERPARAMETERS SETTING

Unless otherwise stated, we use the dataset–backbone specific configurations summarized in Tab. 10. For TUAB and TUEV, models are fine-tuned for 50 epochs with batch size 64 and cosine scheduling: LaBraM uses AdamW with peak learning rate 5e-4, minimal 1e-6, 5 warmup epochs, drop path 0.1, and layer-wise decay 0.65; BIOT uses Adam with peak learning rate 2.5e-3, minimal 1e-6, no warmup, and dropout 0.2. For BCIC2B and PhysioNetP300, we adopt LOSO validation and OneCycle schedules for 100 epochs with batch size 64. BCIC2B starts at 1.5e-5, peaks at 4.0e-4, and decays to 1.5e-7, while PhysioNetP300 starts at 3.0e-5, peaks at 8.0e-4, and decays to 3.0e-7. Other optimizer settings follow Tab. 10 to ensure fair comparison across heads.

# E MORE RESULTS

## E.1 COMPARISONS USING THE CBRAMOD BACKBONE

Table 11: Comparison on TUAB using the CBraMod backbone.

| Methods | Model Size (#Params) | Time / Epoch (m) | Balanced Acc. | Cohen's Kappa | Weighted F1 |
|---|---|---|---|---|---|
| CBraMod (Original)[*] | 4.0M | 10.42 | $0.7973 \pm 0.0024$ | $0.8805 \pm 0.0043$ | $0.8745 \pm 0.0031$ |
| CBraMod + GAP[*] | 4.0M | 10.02 | $0.7978 \pm 0.0031$ | $0.8749 \pm 0.0165$ | $0.8774 \pm 0.0143$ |
| CBraMod + iSQRT-COV | 4.0M+$\Delta$20.3K | 17.89 | $0.8046 \pm 0.0047$ | $0.8873 \pm 0.0064$ | $0.8841 \pm 0.0022$ |
| CBraMod + SVD-Padé | 4.0M+$\Delta$20.3K | 29.97 | $0.8069 \pm 0.0051$ | $0.8910 \pm 0.0053$ | $0.8885 \pm 0.0025$ |
| CBraMod + iSICE | 4.0M+$\Delta$20.3K | 65.38 | $0.8027 \pm 0.0054$ | $0.8858 \pm 0.0049$ | $0.8823 \pm 0.0021$ |
| CBraMod + RHOP | 4.0M+$\Delta$4.6K | 20.14 | $\mathbf{0.8131 \pm 0.0019}$ | $\mathbf{0.8972 \pm 0.0052}$ | $\mathbf{0.8913 \pm 0.0024}$ |

[*] Re-implemented following the official CBraMod preprocessing protocol.

The original CBraMod (Wang et al., 2025a) employs a multi-layer nonlinear classifier rather than a GAP head. To enable a fair comparison, we replaced this classifier with GAP, RHOP, and several existing covariance-based pooling heads, while keeping all remaining components unchanged. Because the exact subject-level data partitions used in CBraMod cannot be faithfully reproduced, we follow the authors' official preprocessing protocol and re-implement the model according to

their recommendations.[5] All models are trained from scratch under the same experimental settings. As shown in Tab. 11, RHOP consistently outperforms both the original classifier and other pooling baselines across all evaluation metrics, highlighting its clear advantage when integrated into the CBraMod backbone.

## E.2 ABLATION ON THE NUMBER OF FRÉCHET MEAN (FM) ITERATIONS

We evaluate the impact of the number of Fréchet Mean (FM) iterations on both TUAB and TUEV. The results are summarized in Tabs. 12 and 13, where the iteration count is set to 1, 2, 3, and $\infty$, with $\infty$ denoting iteration until convergence (i.e., the matrix norm change falling below $1 \times 10^{-5}$).

Across both datasets, performance improves noticeably from 1 to 2 iterations, but the gains saturate quickly thereafter: configurations using 2, 3, or $\infty$ iterations yield nearly identical results. This indicates that a small number of FM iterations is sufficient for stable and effective Riemannian aggregation. Additional iterations incur significant computational overhead without providing further benefits. Consequently, we adopt 1 iteration as the default setting, achieving a strong balance between accuracy and efficiency. This observation is also consistent with prior studies Brooks et al. (2019); Chen et al. (2024b); Chakraborty et al. (2022).

Table 12: Ablation of FM iterations on TUAB.

| Methods | #Iter | Time/Epoch (min) | Balanced Acc. | AUC-PR | AUROC |
|---|---|---|---|---|---|
| LaBraM-Base | 1 | 21.48 | $0.8244 \pm 0.0012$ | $0.9078 \pm 0.0012$ | $0.9105 \pm 0.0011$ |
| LaBraM-Base | 2 | 24.59 | $0.8261 \pm 0.0011$ | $\mathbf{0.9086 \pm 0.0011}$ | $0.9110 \pm 0.0010$ |
| LaBraM-Base | 3 | 27.71 | $\mathbf{0.8263 \pm 0.0012}$ | $0.9085 \pm 0.0011$ | $\mathbf{0.9111 \pm 0.0010}$ |
| LaBraM-Base | $\infty$ | 40.19 | $0.8239 \pm 0.0013$ | $0.9077 \pm 0.0011$ | $0.9107 \pm 0.0010$ |

Table 13: Ablation of FM iterations on TUEV.

| Methods | #Iter | Time/Epoch (min) | Balanced Acc. | Cohen's Kappa | Weighted F1 |
|---|---|---|---|---|---|
| LaBraM-Base | 1 | 2.25 | $0.6380 \pm 0.0056$ | $0.6785 \pm 0.0079$ | $0.8420 \pm 0.0038$ |
| LaBraM-Base | 2 | 3.38 | $\mathbf{0.6401 \pm 0.0054}$ | $0.6802 \pm 0.0077$ | $0.8447 \pm 0.0036$ |
| LaBraM-Base | 3 | 5.02 | $0.6400 \pm 0.0053$ | $\mathbf{0.6803 \pm 0.0076}$ | $\mathbf{0.8451 \pm 0.0037}$ |
| LaBraM-Base | $\infty$ | 10.21 | $0.6360 \pm 0.0054$ | $0.6783 \pm 0.0076$ | $0.8414 \pm 0.0037$ |

## E.3 MORE ABLATION ON $(k, k')$

We further investigate the influence of the embedding dimensions $k$ and $k'$ on model performance. As shown in Tab. 14, in the Quotient-Gaussian embedding, increasing $k$ expands the SPD dimension from $d$ to $d + k$, which amplifies the first-order term $k\mu\mu^{\top}$. As $k$ grows, the embedding becomes more mean-dominated, weakening the relative contribution of second-order statistics that are known to be highly informative for EEG decoding. Moreover, enlarging $d + k$ introduces quadratic growth in matrix size, leading to increased computational burden and potential numerical instability. These effects collectively explain why excessively large $k$ tends to degrade performance.

A parallel interpretation applies to $k'$, which controls the dimensionality of the embedded mean vector: larger values introduce additional first-order information but also increase representation size, resulting in diminishing returns and eventual performance decline. As summarized in Tab. 14, performance improves initially as $(k, k')$ increases, but the gains plateau and then decrease for overly large values. Notably, $(k = 2, k' = 2)$ and $(k = 3, k' = 3)$ consistently lie at or near the optimum across TUAB and TUEV, indicating that moderate dimensionality provides the balance between representational richness, computational efficiency, and numerical stability.

## F FRÉCHET MEAN ON SPD MANIFOLDS

As shown in Alg. 2, the Karcher flow algorithm computes the weighted Fréchet mean (WFM) on the SPD manifold through an iterative process. In each iteration, the data points are projected onto the tangent space at the current estimate $G_{k-1}$ using the logarithmic map, a weighted average is

---

[5]`https://github.com/wjq-learning/CBraMod/blob/main/preprocessing/README.md`

Table 14: Ablation on embedding dimensions $(k, k')$ for LaBraM-Base.

| $k$ | $k'$ | TUAB | | | TUEV | | |
|---|---|---|---|---|---|---|---|
| | | **Balanced Acc.** | **AUC-PR** | **AUROC** | **Balanced Acc.** | **Cohen's Kappa** | **Weighted F1** |
| 1 | 1 | 0.8065±0.0019 | 0.8973±0.0017 | 0.8988±0.0012 | 0.6457±0.0121 | 0.5772±0.0203 | 0.7876±0.0114 |
| | 2 | 0.8193±0.0021 | 0.9054±0.0016 | 0.9064±0.0013 | 0.6339±0.0184 | 0.6078±0.0227 | 0.8062±0.0102 |
| | 3 | 0.8199±0.0020 | 0.9054±0.0015 | 0.9060±0.0012 | **0.6726±0.0137** | 0.6456±0.0181 | 0.8224±0.0097 |
| | 4 | 0.8181±0.0019 | 0.9017±0.0016 | 0.9042±0.0012 | 0.6325±0.0142 | 0.6359±0.0208 | 0.8184±0.0105 |
| 2 | 1 | 0.8171±0.0018 | 0.8975±0.0018 | 0.9013±0.0012 | 0.6631±0.0116 | 0.6471±0.0193 | 0.8252±0.0089 |
| | 2 | **0.8249±0.0022** | 0.9044±0.0014 | 0.9071±0.0011 | 0.6459±0.0128 | 0.6008±0.0251 | 0.7970±0.0095 |
| | 3 | 0.8163±0.0017 | 0.9058±0.0016 | 0.9057±0.0011 | 0.6440±0.0137 | 0.6196±0.0228 | 0.8145±0.0117 |
| | 4 | 0.8164±0.0018 | 0.9043±0.0015 | 0.9046±0.0012 | 0.6409±0.0135 | 0.6302±0.0221 | 0.8129±0.0110 |
| 3 | 1 | 0.8183±0.0019 | 0.9035±0.0015 | 0.9062±0.0011 | 0.6659±0.0123 | 0.6317±0.0273 | 0.8176±0.0135 |
| | 2 | 0.8193±0.0020 | 0.9008±0.0016 | 0.9039±0.0012 | 0.6372±0.0161 | 0.6220±0.0234 | 0.8135±0.0092 |
| | 3 | 0.8244±0.0012 | **0.9078±0.0012** | **0.9105±0.0011** | 0.6380±0.0056 | **0.6785±0.0079** | **0.8420±0.0038** |
| | 4 | 0.8187±0.0021 | 0.9024±0.0017 | 0.9041±0.0013 | 0.6484±0.0138 | 0.6116±0.0217 | 0.8123±0.0109 |
| 4 | 1 | 0.8181±0.0020 | 0.9017±0.0016 | 0.9042±0.0012 | 0.6475±0.0125 | 0.6341±0.0208 | 0.8188±0.0107 |
| | 2 | 0.8164±0.0019 | 0.9043±0.0015 | 0.9046±0.0011 | 0.6362±0.0141 | 0.6193±0.0221 | 0.8127±0.0112 |
| | 3 | 0.8187±0.0021 | 0.9024±0.0017 | 0.9041±0.0013 | 0.6484±0.0138 | 0.6116±0.0217 | 0.8123±0.0109 |
| | 4 | 0.8190±0.0020 | 0.9031±0.0016 | 0.9050±0.0012 | 0.6400±0.0132 | 0.6250±0.0225 | 0.8150±0.0110 |

---

**Algorithm 2:** Karcher Flow Algorithm on the SPD Manifold under AIM

**Input** : A set of SPD matrices $X_{1\dots N} \in \mathcal{S}_d^+$
Number of iterations $K$
**Output** : The FM $G_K \in \mathcal{S}_d^+$
Initialize $G_0 = \frac{1}{N} \sum_{i=1}^{N} X_i$
**for** $k \leftarrow 1$ **to** $K$ **do**
$\quad \mid \quad G_k \leftarrow \mathrm{Exp}_{G_{k-1}} \left( \sum_{i=1}^{N} \mathrm{Log}_{G_{k-1}}(X_i) \right)$
**end**

---

calculated in this tangent space, and the result is mapped back to the manifold using the exponential map. This algorithm is guaranteed to converge on manifolds with non-positive curvatures, such as $\mathcal{S}_d^+$ (Karcher, 1977a). In practice, we initialize $G_0$ with the arithmetic mean and set the number of iterations to $K = 1$, which provides a stable and efficient approximation.

## G LIMITATION

RHOP involves batched SVD to compute matrix logarithms and update sparse inverse covariance. While such operations are generally less optimized than convolutions and attention on current GPUs, in our case, the matrix size is very small (at most $15 \times 15$), making the actual overhead negligible. Thus, this limitation is mainly an implementation detail rather than a practical concern.

## H FUTURE WORK

RHOP is primarily motivated by EEG decoding, but its basic framework and statistics-based principles are general. The Quotient Gaussian Embedding and Riemannian Gaussian Embedding capture two key characteristics that are particularly salient in EEG signals, i.e., spatiotemporal dependency and scale variation. RHOP employs second-order pooling across temporal segments, which preserves fine-grained temporal relations that are lost in first-order pooling schemes. More importantly, the quotient Gaussian provides scale-invariant representations, allowing the model to focus on relational structure rather than absolute amplitude. These theoretical properties make RHOP naturally well-suited for EEG foundation models, which typically rely on mean-based pooling.

However, the RHOP head can also be extended to other multi-channel temporal modalities that exhibit similar structural characteristics, such as ECG, MEG, and fMRI-derived time series. In these domains, the high-order pooling framework may enhance representation ability, while the degree of quotient Gaussian can be adapted to reflect the discriminative role of signal amplitude. Investigating

these directions may help understand when quotient-based modeling is advantageous and shed light on how RHOP could generalize beyond EEG.

## I  Proof of the Thm. 4.2

*Proof of Thm. 4.2* . We construct $\Psi : \mathcal{QN}(n) \to \mathcal{S}_{n+k}^{+,1}$ and show it is a smooth embedding. Fix $k \geq 1$ and define

$$\Phi(C, \mu) = \begin{bmatrix} C + k\mu\mu^\top & \mu^{(k)} \\ \mu^{(k)\top} & I_k \end{bmatrix}, \qquad \mu^{(k)} = \underbrace{[\mu, \dots, \mu]}_{k \text{ columns}}. \tag{14}$$

Observe that $\Phi(C, \mu)$ is SPD and $\det \Phi(C, \mu) = \det C$. Indeed, with

$$T(\mu) = \begin{bmatrix} I_n & \mu^{(k)} \\ 0 & I_k \end{bmatrix}, \tag{15}$$

a direct multiplication gives

$$\Phi(C, \mu) = T(\mu) \begin{bmatrix} C & 0 \\ 0 & I_k \end{bmatrix} T(\mu)^\top. \tag{16}$$

Since $C \in \mathcal{S}_n^+$ and $T(\mu)$ is invertible, $\Phi(C, \mu)$ is SPD. Also $\det T(\mu) = 1$, hence $\det \Phi(C, \mu) = \det C$.

Set $s(C) = (\det C)^{-\frac{1}{n+k}}$ and define

$$\Psi(C, \mu) = s(C)\, \Phi(C, \mu). \tag{17}$$

Then $\det \Psi(C, \mu) = 1$, so $\Psi(C, \mu) \in \mathcal{S}_{n+k}^{+,1}$, matching Eq. (6).

For injectivity and a smooth inverse on the image, suppose $\Psi(C_1, \mu_1) = \Psi(C_2, \mu_2) =: X$, and write $X = \begin{bmatrix} X_{11} & X_{12} \\ X_{21} & X_{22} \end{bmatrix}$. By construction $X_{22} = s(C_i)I_k$, so $s(C_1) = s(C_2) =: s$. Then $X_{12} = s\,\mu_i^{(k)}$ implies $\mu_1 = \mu_2 =: \mu$, and $X_{11} = s(C_i + k\mu\mu^\top)$ yields $C_1 = C_2$.

Conversely, given $X$ in the image, recover $s = \frac{1}{k}\operatorname{tr}(X_{22})$, then $\mu = s^{-1}X_{12}e_1$ (with $e_1$ the first basis vector in $\mathbb{R}^k$), and finally $C = s^{-1}X_{11} - k\mu\mu^\top$. These depend smoothly on $X$ and satisfy $\Psi(C, \mu) = X$. Hence $\Psi$ is a smooth embedding.

It remains to situate this within the affine action. Let $N(n)$ be the space of Gaussians $(\Sigma, \mu)$ with $\Sigma \in \mathcal{S}_n^+$. As shown in Nguyen (2021); Lovrić et al. (2000b), Eqs. (41)–(47), the transitive $Aff^+(n)$-action on $N(n)$ and the embedding $j : Aff^+(n) \hookrightarrow SL(n + k)$ induce

$$(\Sigma, \mu) \longmapsto (\det \Sigma)^{-\frac{1}{n+k}} \begin{bmatrix} \Sigma + k\mu\mu^\top & \mu^{(k)} \\ \mu^{(k)\top} & I_k \end{bmatrix} \in \mathcal{S}_{n+k}^{+,1}. \tag{18}$$

A quotient Gaussian identifies $(\Sigma, \mu)$ up to a positive scalar multiple on $\Sigma$. Choosing the canonical representative $C = (\det \Sigma)^{-\frac{1}{n}}\Sigma$ yields exactly $\Psi(C, \mu)$. Therefore, $\Psi$ is the natural embedding for quotient Gaussians. $\qquad\square$

