# OpenReview forum: "Riemannian High-Order Pooling for Brain Foundation Models"
_ICLR.cc/2026/Conference — ICLR 2026 Poster_

### Official Review · Reviewer_88CQ · 2025-10-17

**Soundness:** 3
**Presentation:** 3
**Contribution:** 3
**Rating:** 8
**Confidence:** 3

**Summary:**

This paper introduces a plug-and-play classification head module named Riemannian High-Order Pooling , specifically designed for Electroencephalography foundation models. The core thesis is that while state-of-the-art EEG foundation models like BIOT and LaBraM have made significant strides in their feature extraction backbones, their classification stages are underdeveloped. These models typically rely on simple pooling mechanisms  or a single class embedding  for prediction, a process that discards second-order statistics and spatiotemporal structure crucial for EEG decoding. RHOP aims to address this deficiency by injecting principled Riemannian statistics and geometric priors into the classifier, thereby making fuller use of the rich features extracted by the foundation model.

**Strengths:**

S1: Significant Problem & Principled Methodology
The paper addresses a well-motivated and significant problem. The proposed solution, RHOP, is not an ad-hoc engineering trick but is built on a solid theoretical foundation.

First, Quotient-Gaussian Embedding  achieves scale invariance. The use of quotient spaces to achieve invariance is a classic approach in geometric machine learning. Applying it to normalize covariance matrices into correlation matrices elegantly addresses the challenge of amplitude variations across EEG segments, a more principled solution than simple normalization techniques.

Second, Riemannian geometry preserves the intrinsic data structure. The long-standing success of Riemannian methods in EEG analysis demonstrates the power of operating on the SPD manifold. By modeling the set of tokens as a Riemannian Gaussian distribution, the method captures not only the central tendency of the features in a geometrically faithful way  but also their dispersion.

Finally, the iSICE layer serves as a feature refiner. Standard covariance captures all pairwise correlations, which can be misleading due to confounding factors. The inverse covariance  captures partial correlations. By using a sparse inverse covariance, iSICE explicitly models and emphasizes the most direct and robust relationships, which is highly beneficial for dealing with noisy EEG data.

These three components are not independent but form a logical, progressive hierarchy of statistical refinement. This structure is a core strength of the method, as it demonstrates a deep understanding of the problem by tackling different statistical challenges at different stages of the pipeline.

S2: Comprehensive and Rigorous Experimental Validation
The experimental evaluation is a major highlight of this paper. The authors have gone to great lengths to demonstrate the effectiveness and generalizability of RHOP.

Diverse Benchmarks: Evaluation on four different and widely-used EEG tasks—abnormality detection on TUAB, event classification on TUEV, motor imagery on BCIC2B, and event-related potentials on PhysioP300—shows that the method is not overfitted to a single problem type. These are standard and challenging benchmarks.

Multiple Backbones: Testing on both BIOT and LaBraM, two state-of-the-art EEG foundation models, proves that RHOP is, as claimed, a truly "backbone-agnostic" module.

Multiple Training Paradigms: The evaluation across full fine-tuning, linear probing, and training from scratch is comprehensive. The strong performance in the "from scratch" and "linear probing" settings is particularly impressive, suggesting that RHOP can both impose useful structure early in training and effectively extract discriminative information from fixed representations.

Strong Baseline Comparisons: The comparison against multiple representative global covariance pooling heads is fair and pits RHOP against its most direct competitors.

S3: Clarity of Exposition & Reproducibility
The paper is well-written, well-structured, and easy to follow. The introduction clearly lays out the motivation. The methods section builds logically from preliminaries to the final framework. Figure 1 provides an excellent visual overview of the entire pipeline. The appendix includes detailed hyperparameters , the Karcher flow algorithm , and a proof for the key embedding theorem , which greatly enhances the work's reproducibility. The authors also promise to release the code upon acceptance, which is in line with academic best practices.

**Weaknesses:**

While the overall methodology is strong, several key design choices lack sufficient justification or analysis.

- Approximation of the Fréchet Mean: The paper states that for efficiency, the Fréchet Mean is computed using only a single iteration of the Karcher flow. While the desire for efficiency is understandable in a deep learning context, this is a significant approximation. The Fréchet Mean is defined as the minimizer of the variance function, and a single step from an arithmetic mean initialization does not guarantee convergence or proximity to the optimum.

- Sensitivity to Hyperparameters ($k, k'$): The embedding dimensions $k$ and $k'$ are crucial hyperparameters.1 Table 6 shows that performance varies with their setting, but no intuition or methodology is provided on how to choose them or set them for a new task. Were they tuned on a validation set? Is there a theoretical reason why $k=3, k'=3$ is optimal on TUAB?

- Choice of Riemannian-Gaussian Embedding (Eq. 9): The specific method for embedding the Riemannian-Gaussian pair $(P^m, P^c)$ into a single SPD matrix $G$ is drawn from the work of Nguyen. While citing prior work is appropriate, a sentence or two explaining why this particular block matrix form involving the Cholesky decomposition of the covariance is a good choice would be highly beneficial to the reader. What properties does this embedding preserve?

**Questions:**

See Weakness

---

> ### Author Response · Authors · 2025-11-28
> **Response to Reviewer 88CQ (1/2)**
>
> We thank Reviewer $\textcolor{orange}{88CQ (R4)}$  for the constructive suggestions and insightful comments! In the following, we respond to the concerns in detail. 😄
>
> ## 1. The Ablation of the Fréchet Mean Iteration.
> Table A: Comparison of TUAB under different FM Iterations.
> |Methods|Number of Iterations|Time / Epoch (min)|Balanced Acc.|AUC-PR|AUROC
> |-|-|-|-|-|-
> |LaBraM-Base|1|21.48|0.8244±0.0012|0.9078±0.0012|0.9105±0.0011
> |LaBraM-Base|2|24.59|0.8261±0.0011|**0.9086±0.0011**|0.9110±0.0010
> |LaBraM-Base|3|27.71|**0.8263 0.0012**|0.9085±0.0011|**0.9111±0.0010**
> |LaBraM-Base|∞|40.19|0.8239±0.0013|0.9077±0.0011|0.9107±0.0010
>
> Table B: Comparison of TUEV under different FM Iterations.
> |Methods|Number of Iterations|Time / Epoch (min)|Balanced Acc.|Cohen’s Kappa|Weighted F1
> |-|-|-|-|-|-|
> |LaBraM-Base|1|2.25|0.6380 ± 0.0056|0.6785 ± 0.0079|0.8420 ± 0.0038
> |LaBraM-Base|2|3.38|**0.6401 ± 0.0054**|0.6802 ± 0.0077|0.8447 ± 0.0036
> |LaBraM-Base|3|5.02|0.6400 ± 0.0053|**0.6803 ± 0.0076**|**0.8451 ± 0.0037**
> |LaBraM-Base|∞|10.21|0.6360 ± 0.0054|0.6783 ± 0.0076|0.8414 ± 0.0037
>
> We report results under different FM iterations (settings as 1, 2, 3, and ∞) in Tables A and B. Wherein, ∞ denotes iterating until convergence, defined as the change in matrix norm falling below $1\times10^{-5}$. Across TUAB and TUEV, performance improves from 1 to 2 iterations but saturates thereafter: 2, 3, and even ∞ iterations yield nearly identical results. This indicates that only a small number of FM iterations is sufficient for stable and effective Riemannian aggregation, whereas additional iterations offer negligible gains while increasing computational cost. Consequently, we adopt 1 iteration, striking a balance between accuracy and efficiency. This choice is also aligned with prior studies [a, b, c].
>
> ## 2. The choices of hyperparameters $(k, k^′)$
> Table C: More ablation of $(k,k')$ on LaBraM-Base.
> |k|k'|TUAB Balanced Acc. |TUAB AUC-PR|TUAB AUROC|TUEV Balanced Acc.|TUEV Cohen’s Kappa|TUEV Weighted F1|
> |-|-|-|-|-|-|-|-|
> |1|1|0.8065±0.0019|0.8973±0.0017|0.8988±0.0012|0.6457±0.0121|0.5772±0.0203|0.7876±0.0114
> |1|2|0.8193±0.0021|0.9054±0.0016|0.9064±0.0013|0.6339±0.0184|0.6078±0.0227|0.8062±0.0102
> |1|3|0.8199±0.0020|0.9054±0.0015|0.9060±0.0012|**0.6726±0.0137**|0.6456±0.0181|0.8224±0.0097
> |1|4|0.8181±0.0019|0.9017±0.0016|0.9042±0.0012|0.6325±0.0142|0.6359±0.0208|0.8184±0.0105
> |2|1|0.8171±0.0018|0.8975±0.0018|0.9013±0.0012|0.6631±0.0116|0.6471±0.0193|0.8252±0.0089
> |2|2|**0.8249±0.0022**|0.9044±0.0014|0.9071±0.0011|0.6459±0.0128|0.6008±0.0251|0.7970±0.0095
> |2|3|0.8163±0.0017|0.9058±0.0016|0.9057±0.0011|0.6440±0.0137|0.6196±0.0228|0.8145±0.0117
> |2|4|0.8164±0.0018|0.9043±0.0015|0.9046±0.0012|0.6409±0.0135|0.6302±0.0221|0.8129±0.0110
> |3|1|0.8183±0.0019|0.9035±0.0015|0.9062±0.0011|0.6659±0.0123|0.6317±0.0273|0.8176±0.0135
> |3|2|0.8193±0.0020|0.9008±0.0016|0.9039±0.0012|0.6372±0.0161|0.6220±0.0234|0.8135±0.0092
> |3|3|0.8244±0.0012|**0.9078±0.0012**|**0.9105±0.0011**|0.6380±0.0056|**0.6785±0.0079**|**0.8420±0.0038**
> |3|4|0.8187±0.0021|0.9024±0.0017|0.9041±0.0013|0.6484±0.0138|0.6116±0.0217|0.8123±0.0109
> |4|1|0.8181±0.0020|0.9017±0.0016|0.9042±0.0012|0.6475±0.0125|0.6341±0.0208|0.8188±0.0107
> |4|2|0.8164±0.0019|0.9043±0.0015|0.9046±0.0011|0.6362±0.0141|0.6193±0.0221|0.8127±0.0112
> |4|3|0.8187±0.0021|0.9024±0.0017|0.9041±0.0013|0.6484±0.0138|0.6116±0.0217|0.8123±0.0109
> |4|4|0.8190±0.0020|0.9031±0.0016|0.9050±0.0012|0.6400±0.0132|0.6250±0.0225|0.8150±0.0110
>
> We agree that the dimensions $k$ and $k'$ have a certain impact on the performance. In all experiments, both hyperparameters were selected using validation sets.
>
> Beyond empirical tuning, there is an intuition for choosing appropriate values. In the Quotient-Gaussian embedding, increasing $k$ enlarges the SPD dimension from $d$ to $d+k$, which scales the first-order term $k\mu\mu^T$ proportionally. This shifts the representation toward mean-dominated statistics and reduces the relative contribution of the second-order statistics, which is known to be more effective for EEG decoding. Hence, small $k$ is qualified to preserve the desired balance, explaining why $k=3$ is sufficient in our experiments. In addition, enlarging $d+k$ introduces quadratic growth in matrix size, increasing computational burden, and numerical instability. These effects degrade accuracy once $k$ becomes larger. A similar interpretation applies to $k'$, which controls the dimensionality of the mean vector during embedding. Larger $k'$ injects more first-order information but increases the dimension, leading again to diminishing returns and eventual performance decline.
>
> As shown in Table C, performance improves as $(k, k')$ increases on TUAB and TUEV, after which the gains plateau and eventually degrade when either dimension becomes larger. Notably, the $(k=2, k'=2)$ and $(k=3, k'=3)$ consistently lie at or near the optimum, indicating that moderate dimensionality provides the best trade-off between expressiveness and stability.

---

> > ### Author Response · Authors · 2025-11-28
> > **Response to Reviewer 88CQ (2/2)**
> >
> > ## 3. The benefits of Riemannian Gaussian embedding
> >
> > As shown in [a, Def. 3 and Thm. 1], the Riemannian Gaussian pair $(P^{m}, P^{c})$ lies on a product SPD manifold $\mathcal{M}(n,n')=\mathcal{S}^+_n \times \mathcal{S}^+_{n'}$, which becomes a Lie group under the following group
> > $$
> > (P^{m}_1,P^{c}_1)\star(P^{m}_2,P^{c}_2)(
> > \varphi^{-1}(\varphi(P^m_1)L_2+\varphi(P^{m}_2),(L_1L_2)(L_1L_2)^T),
> > $$
> > where $P^{c}_i=L_i L_i^T$ is the Cholesky factor, and $\varphi$ is a smooth bijection. In other words, the Riemannian mean and covariance are not just two unrelated SPD matrices, but form a point on a Lie group with well-defined group multiplication and inverse.
> >
> > Furthermore, [a, Thm. 2] proves that the proposed block-matrix embedding is a Lie-group isomorphism. A mapping $\Phi : G_1 \to G_2$ between two Lie groups is a Lie-group isomorphism if it satisfies the following conditions:
> > - It preserves group structure $$\Phi(g_1 \circ g_2)=\Phi(g_1)\bullet\Phi(g_2), \forall g_1, g_2 \in G_1.$$
> > - It is smooth and invertible, and the inverse is also smooth.
> > The Cholesky-based block structure ensures the resulting matrix is always SPD and differentiable, making the representation numerically stable and fully compatible with end-to-end optimization.
> >
> >
> > **References**
> >
> > [a] GeomNet: A Neural Network Based on Riemannian Geometries of SPD Matrix Space and Cholesky Space for 3D Skeleton-Based Interaction Recognition.
> >
> > [b] Riemannian batch normalization for SPD neural networks
> >
> > [c] SPD domain-specific batch normalization to crack interpretable unsupervised domain adaptation in EEG.

---

### Official Review · Reviewer_sQZN · 2025-10-19

**Soundness:** 2
**Presentation:** 3
**Contribution:** 2
**Rating:** 4
**Confidence:** 4

**Summary:**

This paper presents an exploratory attempt to integrate recent large brain foundation models with second-order pooling techniques originally developed in computer vision. The authors claim that their proposed neural layer captures second-order information and global spatiotemporal dependencies from EEGs, and evaluate their approaches on several EEG-BCI scenarios.

**Strengths:**

Overall, the work is conceptually interesting. It aims to transfer second-order modeling paradigms from vision to EEG-based brain signal analysis. The motivation is creative, and such cross-domain adaptation deserves attention.

**Weaknesses:**

1. The paper frequently cites SPDNet approaches where SPD matrices are computed as spatial covariance matrices derived from multichannel EEG time series. However, the current work constructs SPD representations from tokens derived from the backbone model on temporal segment EEGs, i.e., covariances across temporal-segment tokens rather than across spatial channels. This formulation deviates from existing literature on Riemannian EEG analysis, where spatial covariance carries physiologically meaningful information about inter-channel connectivity. Since reaction time and task onset vary substantially across trials, temporal alignment is inconsistent, making temporal-segment covariance unlikely to encode stable or discriminative features. The marginal performance gains across datasets and backbones (Tables 1–4) further support this concern. Without a clear justification, the connection between the proposed SPD representation and neural dynamics remains weak.

2. The experimental setup is under-specified. It is unclear whether the experiments are subject-independent or subject-dependent, how many cross-validation folds were used, and what preprocessing pipeline (e.g., filtering, artifact rejection, normalization) was applied. These details are critical, as EEG performance is highly sensitive to preprocessing and evaluation protocols. Moreover, the description of the BCIC2B dataset is ambiguous; it is not stated which BCI Competition dataset (e.g., IV-2a, II-2b, etc.) was used. Given that such datasets are relatively small, large-scale models like LaBraM-Base are highly prone to overfitting. Without explicit regularization or validation strategies, the reported performance may not generalize.

**Questions:**

1. What is the physiological or theoretical motivation for the covariance of the temporal-segment tokens to carry class-discriminative information? Has any prior literature demonstrated its effectiveness for the EEG-BCI classification?

2. Can the authors clarify the experimental protocol in detail? Were the experiments subject-dependent or subject-independent? How many folds or runs were used for cross-validation? What preprocessing steps were applied to the EEG signals (e.g., band-pass filtering, normalization, artifact removal)?

3. Please specify all experimental settings in all experiments. Regarding the "BCIC2B" dataset, please specify: Which exact dataset version or competition subset was used (e.g., BCI Competition IV-2a or II-2b)? Provide a download link or public reference for reproducibility. What measures were taken to mitigate overfitting, such as early stopping, data augmentation, dropout, or weight decay?

4. Although the EEG-BCI classification inspires the proposed architecture, its design, particularly the covariance of the temporal-segment token and Riemannian high-order pooling, seems more general and not specific to EEG. Could the authors discuss whether their model can be effectively applied to other domains where temporal dependencies and non-Euclidean structure exist (e.g., video, speech, or physiological time series)? If so, what properties of those modalities would make the approach more suitable or interpretable than in EEGs?

---

> ### Author Response · Authors · 2025-11-28
> **Response to Reviewer sQZN (1/3)**
>
> We thank Reviewer $\textcolor{purple}{sQZN (R3)}$ for the careful review and the suggestive comments. Below, we address the comments in detail. 😄
>
> ## 1. Theoretical motivation and novelty of using temporal-token covariance in EEG decoding.
>
>
> Table A: Gaussianity statistics of GCP token-level aggregated features. Here, $p>0.05$ indicates non-rejection of Gaussianity.
> |Model|Dataset|Mean Shapiro-Wilk p-value|Proportion (p > 0.05)|Mean skewness|Mean excess kurtosis|
> |-|-|-|-|-|-|
> |LaBraM|TUAB|0.01|0.05|0.14|18.94|
> |LaBraM|TUEV|0.01|0.03|0.25|19.54|
> |BIOT|TUAB|0.02|0.03|0.22|19.16|
> |BIOT|TUEV|0.01|0.00|0.14|19.97|
>
> Table B: Gaussianity statistics of RHOP temporal-segment features. Here, $p>0.05$ indicates non-rejection of Gaussianity.
> |Model|Dataset|Mean Shapiro-Wilk p-value|Proportion (p > 0.05)|Mean skewness|Mean excess kurtosis|
> |-|-|-|-|-|-|
> |LaBraM|TUAB|0.49|0.94|-0.13|0.19|
> |LaBraM|TUEV|0.47|0.89|-0.10|0.15|
> |BIOT|TUAB|0.46|0.91|-0.07|0.10|
> |BIOT|TUEV|0.44|0.85|-0.09|0.12|
>
>
> Our theoretical motivation builds directly on prior studies in global covariance pooling (GCP) [a, b, c], which demonstrate that second-order statistics of high-level activations contain discriminative information complementary to first-order statistics. Specifically, GCP summarizes a set of feature vectors $z_1,\ldots,z_N$ by the covariance
> $$
> \Sigma_{\mathrm{GCP}} = \frac{1}{N-1} \sum_{n=1}^N (z_n - \bar{z})(z_n - \bar{z})^\top \in \mathbb{R}^{N \times N},
> $$
> capturing second-order relationships among latent feature dimensions and yielding a richer representation than mean pooling. However, from a statistical perspective, directly applying GCP to the token dimension of EEG representations is inappropriate.
>
>
> - **Gaussianity of GCP token-level features.** As shown in Table A, we conduct a random-direction Gaussianity test on the learned token features and observe strong deviation from Gaussianity, reflected by large excess kurtosis and skewed heavy-tailed behavior. This non-Gaussian structure is likely induced by the complex temporal–spatial dynamics of EEG, where mixing heterogeneous spatial and temporal components leads to multimodal and heavy-tailed distributions. Since covariance is not a sufficient statistic for non-Gaussian data, aggregating features across the token dimension produces unstable descriptors.
>
> - **Gaussianity of RHOP temporal-segment features.** As shown in Table B,when examining the temporal-segment features, the Gaussianity tests reveal unimodal and approximately Gaussian distributions, with moderate Shapiro–Wilk $p$-values, low skewness, and substantially smaller kurtosis. This makes temporal segments the appropriate unit for second-order modeling. Therefore, instead of computing covariance across tokens as in GCP, we compute covariance within temporal segments, which yields a statistically meaningful estimator.
>
> - **Novelty of temporal-segment covariance.** To the best of our knowledge,  no existing EEG-BCI work applies GCP or second-order token covariance to temporal embeddings. Classical Riemannian EEG pipelines operate on the spatial covariance of raw EEG, while most deep-learning-based EEG decoders rely on global average pooling (GAP) or token concatenation, thereby discarding global second-order information. This gap is precisely the motivation behind Riemannian high-order pooling (RHOP). Our contribution is to extend the proven effectiveness of second-order representation learning to the spatiotemporal token representations generated by modern EEG foundation models such as BIOT and LaBraM. By introducing a quotient-Gaussian embedding that normalizes per-token covariances and a Riemannian Gaussian embedding that aggregates across tokens on the SPD manifold, RHOP preserves the spatiotemporal structure inherent to EEG while capturing high-order dependencies that standard GCP and GAP cannot exploit.
> - **Regarding temporal misalignment.** RHOP does not operate on raw EEG signals. Instead, our inputs are higher-level semantic embeddings produced by EEG foundation models such as BIOT and LaBraM. These models already incorporate substantial temporal alignment mechanisms through stacked spatiotemporal convolutions, self-attention layers, and multiple Transformer blocks, which explicitly model temporal dependencies and normalize variations in trial onset. As a result, the token sequences feeding into RHOP are temporally stabilized representations that capture task-relevant structure rather than raw, unaligned sensor readings.

---

> ### Author Response · Authors · 2025-11-28
> **Response to Reviewer sQZN (2/3)**
>
> ## 2. On the evaluation protocol and preprocessing details.
>
> We clarify the full experimental protocol used in our work.
>
> - **Subject independence.** All experiments in the paper adopt a *strict subject-independent* evaluation protocol. For datasets that provide an official subject-wise split (e.g., TUAB, TUEV), we follow the official training–test separation and further divide the training set into an 80–20 train–validation split, consistent with prior EEG foundation model work i.e. BIOT and LaBraM. For BCIC2B, PhysioP300 datasets, we use a Leave-One-Subject-Out (LOSO) cross-validation protocol. Each fold uses one subject for testing and all remaining subjects for training.
>
> - **Number of folds / runs.** For LOSO datasets (BCIC2B, PhysioP300), this corresponds to **9-fold** cross-validation. For TUAB and TUEV, the number of subjects follows the official partitions. To improve statistical reliability, all experiments are repeated 3 times with different random seeds, and we report mean ± standard deviation.
>
> - **Preprocessing steps.** We provide dataset-specific preprocessing to match prior work:
>   - **TUAB and TUEV.** EEGs are first resampled to 200 Hz. For BIOT-based models, we apply the *16-channel bipolar montage* used in BIOT. For LaBraM-based models, we use LaBraM’s minimal preprocessing pipeline (0.1–75 Hz band-pass + 50 Hz notch + unit normalization).
>   - **BCIC2B.** We remove all EOG channels. EEG is band-pass filtered to 0–38 Hz, resampled to 200 Hz, and normalized using EA normalization [d].
>   - **PhysioNetP300.** We retain all 64 channels. Signals are filtered to 0–120 Hz and resampled to 200 Hz. Each trial is epoched to a 2-second window aligned to stimulus onset (−0.1 s to 2 s). [e].
>
> These preprocessing steps align consistently with the EEG foundation models on which our baselines are built, while maintaining fairness across all experimental settings.
>
> ## 3. Clarifications on Dataset Choice and Regularization for BCIC2B.
>
> Additional hyperparameters and implementation details for all datasets are provided in Appendices C and  D.
>
> For BCIC2B, we use the **BCI Competition IV – Dataset 2b**
> ([https://www.bbci.de/competition/iv/#datasets](https://www.bbci.de/competition/iv/#datasets)).
> All experiments adopt the same **subject-independent** LOSO protocol already described in Q2, with **9 folds** and **three repeated runs** per fold. The preprocessing is summarized in Q2.
>
>
> To mitigate overfitting, we employ several regularization mechanisms:
>
> * **Weight decay $(5×10^{-2})$** is deliberately set to a relatively large value to penalize high-magnitude parameters.
> * We incorporate iSICE to regularize the covariance representation. iSICE solves an L1-regularized precision-matrix estimation problem to obtain a sparse inverse covariance. This removes indirect or confounding dependencies, yielding a noise-robust and parsimonious dependency structure that prevents overfitting to spurious inter-channel correlations.
> * We adopt **linear probing** by freezing all pretrained LaBraM transformer blocks and training only the lightweight classifier layers. This reduces the number of trainable parameters by more than an order of magnitude (≈5.8M → 0.5K), making the model far less prone to overfitting.
> * We additionally set a drop-path rate of 0.1, providing mild stochastic regularization within the pretrained transformer blocks.

---

> > ### Author Response · Authors · 2025-11-28
> > **Response to Reviewer sQZN (3/3)**
> >
> > ## 4. **Scope and Applicability of RHOP Beyond EEG**
> >
> >
> > Although RHOP is motivated by EEG decoding, its key components, i.e., Quotient Gaussian Embedding and Riemannian Gaussian Embedding, are built on properties that are particularly prominent in EEG:
> > * **Preserving spatiotemporal structure.** EEG exhibits strong spatiotemporal structures. To preserve these dependencies, RHOP performs second-order pooling across temporal-segment tokens rather than collapsing all tokens as in standard GCP.
> > * **Scale-invariant.** EEG signals also show substantial amplitude and scale variation across segments. The Quotient Gaussian Embedding provides scale-invariant normalization, ensuring that representations focus on dependency structure rather than absolute magnitude, which is more informative for EEG tasks. These properties make RHOP particularly suitable for EEG foundation models, which currently rely almost exclusively on first-order pooling.
> >
> > Given these considerations, RHOP is not restricted to EEG. Other modalities with multi-channel temporal dynamics and structured spatiotemporal dependencies, such as ECG, MEG, or fMRI-derived time series, can benefit from the same high-order pooling framework. However, these modalities may not require the scale-invariant quotient embedding, since signal amplitude can itself be discriminative; a standard Gaussian embedding may be sufficient. In contrast, domains such as video or speech provide weaker justification: their latent features are typically less SPD-structured, and Riemannian covariance modeling has not been widely adopted in those areas.
> >
> >
> >
> > **References**
> >
> > [a] Is Second-order Information Helpful for Large-scale Visual Recognition?
> >
> > [b] Deep CNNs meet global covariance pooling: Better representation and generalization.
> >
> > [c] Sot: Delving deeper into classification head for transformer.
> >
> > [d] Transfer learning for brain–computer interfaces: A Euclidean space data alignment approach.
> >
> > [e] BENDR: Using transformers and a contrastive self-supervised learning task to learn from massive amounts of EEG data.

---

### Official Review · Reviewer_VvTH · 2025-10-29

**Soundness:** 2
**Presentation:** 3
**Contribution:** 2
**Rating:** 4
**Confidence:** 4

**Summary:**

This paper proposes RHOP, a pooling module for EEG foundation models that uses quotient Gaussian embeddings and Riemannian aggregation on SPD manifolds. While the geometric approach is conceptually interesting, the paper suffers from multiple critical flaws: missing the current SOTA baseline (CBraMod in ICLR'25), potential rank deficiency issues, unjustified Gaussian assumptions, and marginal empirical gains.

**Strengths:**

The method is conceptually interesting and represents a novel approach to token-level aggregation in EEG foundation models. The paper is well-written with clear methodology and good presentation.

**Weaknesses:**

1.Missing Backbone (CBraMod in ICLR'25):

The paper cites CBraMod [1] in page 2 but did not evaluate on it. CBraMod outperforms LaBraM, BIOT, and other EEG foundation models, as well as backbone + RHOP proposed in this work, while being substantially lighter in terms of trainable params. Moreover, CBraMod[1] is open‑source and provides pretrained checkpoints, which makes it directly compatible as a plug‑in backbone for RHOP. Ignoring CbraMod as backbone limits the paper's contribution.

2. Rank Deficiency Not Resolved:

The paper does not clearly specify the values of $D$, $N$, $T$, used by each baseline on each dataset. This prevents proper assessment of issues such as rank-deficiency.
When \( D < T \) (common in EEG), the covariance matrix \( $\Sigma_n \in \mathbb{R}^{T \times T}$ \) is rank-deficient, with
rank($\Sigma_n) \leq D $.
$C_n = D_n^{-1/2} \Sigma_n D_n^{-1/2}\$ preserves rank and therefore cannot convert a semi-definite matrix into a strictly positive definite (SPD) one.
Moreover, in Equation (11), $\mu_n^{(k)} (\mu_n^{(k)})^\top = k \mu_n \mu_n^\top$, leaving the Schur complement as
$S = C_n$. This will leave $Y_n$ in Eq. 11 not SPD, therefore, Riemannian geometry should not be used.
This severe rank deficiency fundamentally limits the informativeness of the SPD representation.
Particularly, In BCIC2B, where $D=3$, the resulting covariance matrix has very low rank, limiting its capability to capture informative SPD structure.

3. Gaussian Assumption Unjustified:

RHOP applies Quotient‑Gaussian embedding to learned token features, not to raw EEG. However, these features are outputs of deep encoders (LaBraM or BIOT), including non-linear activations, normalization layers, and attention mechanisms, making Gaussian distribution highly unlikely, leaving the SPD embedding theoretically questionable. Covariance is not a sufficient statistic for non-Gaussian data, and QGE may discard essential structure.

4. Across several datasets, the improvement of baseline + RHOP over the corresponding baseline is marginal (often less than 1%).

Refs:

[1] Wang, Jiquan, et al. "Cbramod: A criss-cross brain foundation model for eeg decoding." arXiv preprint arXiv:2412.07236 (2024).


I would be willing to raise my score if the authors address any concerns outlined above.

**Questions:**

See comments above.

---

> ### Author Response · Authors · 2025-11-28
> **Response to Reviewer VvTH (1/3)**
>
> We thank Reviewer $\textcolor{red}{VvTH (R2)}$ for the careful review and the suggestive comments. Below, we address the comments in detail. 😄
>
> ## 1. Comparisons of the CBraMod Backbone
>
> Table A: Comparison of TUAB based on CBraMod Backbone.
> |Methods|Balanced Acc.|AUC-PR|AUROC|
> |-|-|-|-|
> |CBraMod(Original)|0.7973 ± 0.0024|0.8805 ± 0.043|0.8745 ± 0.0031|
> |CBraMod+GAP|0.7978 ± 0.0031|0.8749 ± 0.0165|0.8774 ± 0.0143|
> |CBraMod+iSQRT-COV|0.8046 ± 0.0047|0.8873 ± 0.064|0.8841 ± 0.0022|
> |CBraMod+SVD-Padé|0.8069 ± 0.0051|0.8910 ± 0.053|0.8885 ± 0.0025|
> |CBraMod+iSICE|0.8027 ± 0.0054|0.8858 ± 0.049|0.8823 ± 0.0021|
> |CBraMod+RHOP|**0.8131 ± 0.0019**|**0.8972 ± 0.0052**|**0.8913 ± 0.0024**|
>
>
>
> The original CBraMod uses a multi-layer nonlinear classifier rather than a Global Average Pooling (GAP) head. For fair comparison, we replaced this classifier with GAP, RHOP, or other GCP heads while keeping all other components unchanged.  Since the exact data partitions used in CBraMod cannot be reliably reproduced, we followed the authors’ recommendation to reproduce the results. More details can be found in: <https://github.com/wjq-learning/CBraMod/blob/main/preprocessing/README.md>. As shown Table A, RHOP consistently outperforms both the original CBraMod classifier and the GAP across all metrics, demonstrating its clear advantage.
>
> ## 2. Preventing Rank Deficiency in Covariance Estimation
>
> Table B: The number of channels and temporal segment Dimensions.
> |Dataset|Number of channel|Number of temporal segments|
> |-|-|-|
> TUAB|23|10|
> TUEV|23|5|
> BCIC2B|3|4|
> PhysioP300|64|2|
>
>
> - **The dimension of $D$ and $T$.** In EEG foundation models, such as LaBraM and BIOT, data are processed as temporal-slice tokens. For TUAB, the preprocessed input is: $X \in \mathbb{R}^{16 \times (10 \cdot 200)}$, which is reshaped into: $X' \in \mathbb{R}^{16 \times 10 \times 200}$. Here, the temporal dimension used for covariance estimation is the number of temporal segments: $T=10$ instead of the raw temporal resolution (e.g., 2000 time points). As summarized in Table B, across all datasets, we typically observe $T < D$, and even in BCIC2B, $D$ and $T$ are comparable rather than in a degenerate regime.
>
> - **Ensuring strict positive definiteness.** Additionally, to guarantee a strictly positive definite (SPD) matrix  before embedding, we apply a regularization
> $$
> \Sigma_n \leftarrow \Sigma_n + \epsilon I.
> $$
> This practice is standard in SPD manifold–based EEG decoding and ensures $\Sigma_n \succ 0$. The correlation mtrix $C_n = D_n^{-1/2},\Sigma_n,D_n^{-1/2}$ is also SPD. Consequently, the Schur complement remains positive definite, ensuring that the quotient Gaussian embedding is a SPD matrix.

---

> ### Author Response · Authors · 2025-11-28
> **Response to Reviewer VvTH (2/3)**
>
> ## 3. Empirical Assessment of the Gaussianity Assumption
>
> Table C: Gaussianity statistics of RHOP temporal-segment features. Here, $p>0.05$ indicates non-rejection of Gaussianity.
> |Model|Dataset|Mean Shapiro-Wilk p-value|Proportion (p > 0.05)|Mean skewness|Mean excess kurtosis|
> |-|-|-|-|-|-|
> |LaBraM|TUAB|0.49|0.94|-0.13|0.19|
> |LaBraM|TUEV|0.47|0.89|-0.10|0.15|
> |BIOT|TUAB|0.46|0.91|-0.07|0.10|
> |BIOT|TUEV|0.44|0.85|-0.09|0.12|
>
> Table D: Gaussianity statistics of GCP token-level aggregated features. Here, $p>0.05$ indicates non-rejection of Gaussianity.
> |Model|Dataset|Mean Shapiro-Wilk p-value|Proportion (p > 0.05)|Mean skewness|Mean excess kurtosis|
> |-|-|-|-|-|-|
> |LaBraM|TUAB|0.01|0.05|0.14|18.94|
> |LaBraM|TUEV|0.01|0.03|0.25|19.54|
> |BIOT|TUAB|0.02|0.03|0.22|19.16|
> |BIOT|TUEV|0.01|0.00|0.14|19.97|
>
>
> - **Quotient Gaussian embedding is a statistical representation.** Our quotient-Gaussian embedding jointly encodes the mean and scale-invariant covariance, i.e., second-order statistics, whereas Global Average Pooling (GAP) only keeps first-order statistics, i.e., means. Although covariance is not a sufficient statistic for general non-Gaussian data, it still captures second-order information as a complement. Importantly, prior GCP methods operate on deep features and still achieve strong gains. indicating that exploiting second-order structure is beneficial, although feature distributions slightly deviate from Gaussian.
>
> - **Gaussianity of RHOP temporal-segment features.** To assess the distributions of features, we apply a random-direction test by projecting the temporal-dimensional token features onto multiple random 1-D directions. For each projection, we compute the Shapiro–Wilk p-value, skewness, and excess kurtosis, and finally average these statistics over all batches. As summarized in Table C, the projected features exhibit **high Shapiro–Wilk p-values (0.44–0.49)** and **very small skewness**, indicating approximately symmetric and unimodal distributions. The excess kurtosis is slightly positive, reflecting mild heavy-tailedness. All in all, the distributions remain **close to Gaussian** across all datasets and backbones.
>
> - **Gaussianity of GCP token-level features.** In contrast, as shown in Table C, GCP evaluates Gaussianity along the token dimension, where spatial and temporal components are mixed. As shown in Table D, this axis exhibits near-zero Shapiro–Wilk $p$-values ($0.00$–$0.02$), fewer than $5\%$ of projections with $p>0.05$, and extremely large excess kurtosis ($18.9$–$19.9$), indicating strongly heavy-tailed and non-Gaussian behavior. Covariance computed on such mixed representations is therefore statistically unstable.
>
> These results show that RHOP operates on statistically well-behaved temporal-segment features, whereas GCP aggregates incompatible spatial–temporal components. RHOP’s temporal axis admits stable first- and second-order statistics, making the quotient-Gaussian embedding more coherent with the data structure and likely contributing to its stronger EEG decoding performance.

---

> ### Author Response · Authors · 2025-11-28
> **Response to Reviewer VvTH (3/3)**
>
> ## 4. Performance Improvements of RHOP
>
> We computed the absolute performance gains of RHOP over all compared pooling heads (BIOT/LaBraM backbones and all GCP variants: iSQRT-COV, SVD-Padé, and iSICE) across all datasets and metrics. Among these 92 improvements, **65 exceed +1%**, accounting for **70.7% of all comparisons**. This includes many substantial improvements, such as:
> * **+10.08%** (TUEV BIOT non-pretrained, SVD-Padé → RHOP, Balanced Acc.)
> * **+16.33%** (TUEV BIOT non-pretrained, iSQRT-COV → RHOP, Kappa)
>
> When comparing only against backbone models (i.e., BIOT vs BIOT+RHOP or LaBraM vs LaBraM+RHOP), RHOP still achieves consistently meaningful gains. Out of 24 backbone-level comparisons, **13 exceed +1%**, corresponding to **56.5% of all backbone improvements**. Representative examples include:
> * **+6.73%** on TUEV (BIOT non-pretrained, Balanced Acc.),
> * **+2.91%** on TUEV (BIOT pretrained, Balanced Acc.),
> * **+1.04%** on TUAB (LaBraM pretrained, Balanced Acc.),
>
> These results demonstrate that the gains of RHOP are far from marginal. They reflect consistent improvements across datasets, backbones, and evaluation metrics, confirming that RHOP provides meaningful advantages over both standard backbone pooling and advanced GCP-based methods.
>
>
> Table E: RHOP performance gains (Δ) on TUEV — BIOT (non-pretrained).
> |Method|Balanced Acc. (Δ)|Cohen’s Kappa (Δ)|Weighted F1 (Δ)|
> |-|-|-|-|
> |BIOT|+0.0673|+0.0695|+0.0381|
> |iSQRT-COV|+0.0875|**+0.1633**|**+0.1143**|
> |SVD-Padé|**+0.1008**|+0.1086|+0.0560|
> |iSICE|+0.0725|+0.0614|+0.0326|
>
> Table F: RHOP performance gains (Δ) on TUEV — BIOT (pretrained).
> |Method|Balanced Acc. (Δ)|Cohen’s Kappa (Δ)|Weighted F1 (Δ)|
> |-|-|-|-|
> |BIOT|+0.0291|+0.0187|+0.0073|
> |iSQRT-COV|+0.0889|**+0.1897**|**+0.1035**|
> |SVD-Padé|**+0.1200**|+0.0634|+0.0166|
> |iSICE|+0.0214|+0.0215|+0.0031|
>
>
> Table H: RHOP performance gains (Δ) on TUEV — LaBraM-Base (pretrained).
> |Method|Balanced Acc. (Δ)|Cohen’s Kappa (Δ)|Weighted F1 (Δ)|
> |-|-|-|-|
> |LaBraM|−0.0029|+0.0148|+0.0108|
> |iSQRT-COV|+0.0144|+0.0638|+0.0358|
> |SVD-Padé|**+0.0775**|**+0.0987**|**+0.0520**|
> |iSICE|−0.0025|+0.0651|+0.0238|
>
>
> Table I: RHOP performance gains (Δ) on TUAB — BIOT (non-pretrained).
> |Method|Balanced Acc. (Δ)|Cohen’s Kappa (Δ)|Weighted F1 (Δ)|
> |-|-|-|-|
> |BIOT|+0.0068|+0.0012|+0.0074|
> |iSQRT-COV|+0.0010|+0.0035|+0.0106|
> |SVD-Padé|**+0.0490**|**+0.0445**|**+0.0495**|
> |iSICE|+0.0034|−0.0073|−0.0050|
>
>
> Table J: RHOP performance gains (Δ) on TUAB — BIOT (pretrained).
> |Method|Balanced Acc. (Δ)|AUC-PR (Δ)|AUROC (Δ)|
> |-|-|-|-|
> |BIOT|+0.0143|+0.0041|+0.0049|
> |iSQRT-COV|+0.0283|+0.0243|+0.0266|
> |SVD-Padé|**+0.0570**|**+0.0559**|**+0.0594**|
> |iSICE|+0.0126|+0.0216|+0.0125|
>
> Table K: RHOP performance gains (Δ) on TUAB — LaBraM-Base (pretrained).
> |Method|Balanced Acc. (Δ)|AUC-PR (Δ)|AUROC (Δ)|
> |-|-|-|-|
> |LaBraM|**+0.0104**|**+0.0113**|**+0.0083**|
> |iSQRT-COV|+0.0056|+0.0039|+0.0045|
> |SVD-Padé|+0.0042|+0.0016|+0.0033|
> |iSICE|+0.0061|+0.0041|+0.0046|
>
>
> Table L: RHOP performance gains (Δ) on BCIC2B.
> |Method|Balanced Acc. (Δ)|AUC-PR (Δ)|AUROC (Δ)|
> |-|-|-|-|
> |LaBraM-Base|+0.0061|+0.0080|+0.0115|
> |iSQRT-COV|+0.0259|+0.0275|**+0.0302**|
> |SVD-Padé|**+0.0272**|**+0.0287**|+0.0320|
> |iSICE|+0.0030|+0.0052|+0.0077|
>
>
> Table M: RHOP performance gains (Δ) on PhysioP300.
> |Method|Balanced Acc. (Δ)|AUC-PR (Δ)|AUROC (Δ)|
> |-|-|-|-|
> |LaBraM-Base|+0.0190|+0.0065|+0.0151|
> |iSQRT-COV|+0.0347|+0.0070|+0.0231|
> |SVD-Padé|+0.0194|+0.0230|+0.0293|
> |iSICE|**+0.0379**|**+0.0280**|**+0.0617**|

---

### Official Review · Reviewer_Pf1i · 2025-11-01

**Soundness:** 4
**Presentation:** 3
**Contribution:** 3
**Rating:** 8
**Confidence:** 2

**Summary:**

The paper highlights a shortcoming of existing foundation models trained on EEG signals: they ignore the informative geometry of the underlying spaces. The paper addresses this by proposing a simple module that can be added to (even pre-trained) existing foundations models to allow them to make use of Riemannian geometry, which is well-known in the BCI community to capture important task-relevant information.

**Strengths:**

- comprehensive, compelling experiments
- open source code and applications on public datasets
- well written
- clear theoretical motivation
- nice translation of the theory into a practical method

**Weaknesses:**

- the clear performance gains of the method are nevertheless modest
- the focus is a bit limited (compared to the broader ICLR community), focusing on EEG applications

**Questions:**

1. are there other applications of this besides EEG?
2. L058: Is it really about scaling, or more about applying/tailoring/tuning?
3. L070: What does CLS stand for?
4. L089 and 094: SPD is an adjective, so what's the missing noun it's modifying here? Matrices? Manifold?
5. L104 and 106: A word is missing after EEG here. Maybe "signals" or "measurements"?
6. L127: Can the authors more precisely and mathematically explain what is meant here (and elsewhere in the section) by "collapses all spatiotemporal tokens into a single global discriptor"? Isn't the proprosed method also doing that via eq. (9), but in a geometry-aware way?
7. L185 and 189: $\mathcal{Q}\mathcal{N}(n)$ isn't a single distribution but the space of all distributions, isn't it? And then eq. (5) is a specific (quotient) distribution that is also a representative of the equivalence class of the nonquotient distributions? So each element $[\Sigma, \mu]$ is an equivalence class of distributions $(\Sigma^\prime, \mu)$ where ${\Sigma}, {\Sigma^\prime}$ have the same corresponding correlation matrix?
8. L194: why not use something more informative and accessible than "↓", like "Appendix F"?
9. L210: is eq. (7) here the same as (4)?
10. Tables 3 and 4: why does the param count for LaBraM-Base+iSICE drop by over 20k here, while all other methods have little or no drop?

---

> ### Author Response · Authors · 2025-11-28
> **Response to Reviewer Pf1i (1/2)**
>
> We thank Reviewer $\textcolor{blue}{\rm{Pf1i (R1)}}$ for the careful review and the suggestive comments. Below, we address the comments in detail. 😊
>
>
> ## 1 (Q1). Scope and Applicability of RHOP Beyond EEG
>
> We focus on EEG signals for three reasons:
> * **Effectiveness of SPD manifold methods.** Prior works have shown that SPD manifold methods are effective for EEG decoding [a, b, c]. This is because they are especially well-suited for EEG data: SPD representations are invariant to linear mixing of latent neural sources [d], and provide consistent [e] and intrinsically interpretable [f] estimators for generative models in which class information is encoded through log-linear modulations of source power.
> * **Preserving spatiotemporal structure.** EEG exhibits strong spatiotemporal dependencies. Standard GCP heads [g, h, i] ignore the intrinsic spatiotemporal structure of EEG features. To preserve these dependencies, RHOP performs second-order pooling across temporal-segment tokens rather than collapsing all tokens as in standard GCP.
> * **Scale-invariant.** EEG signals also show substantial amplitude and scale variation across segments. The Quotient Gaussian Embedding provides scale-invariant normalization, ensuring that representations focus on dependency structure rather than absolute magnitude, which is often more informative for EEG tasks. These properties make second-order, geometry-aware pooling particularly suitable for EEG foundation models, which currently rely almost exclusively on first-order pooling.
>
>
> Nonetheless, this limitation of global covariance pooling (GCP) also appears in other multi-channel temporal signals. Thus, RHOP is not restricted to EEG and can be applied to electrocardiography (ECG), magnetoencephalography (MEG), and fMRI-derived time series. We plan to explore these extensions in future work.
>
> ## 2. Minor wording issues: Q2–Q5, Q8, Q10
>
> - **Q2 (L058).** We mean applying the backbone to downstream tasks, not scaling. We have replaced “scaling’’ with “applying’’.
> - **Q3 (L070).** CLS refers to the classification token, following transformer terminology. We now explicitly spell out “classification (CLS) token’’ at first mention.
> - **Q4 (L089 & L094).** In both cases, “SPD’’ should modify a noun. We have revised the text to “the SPD manifold.’’
> - **Q5 (L104 & L106).** We added the missing nouns and now use “EEG signals’’ and “EEG features.’’
> - **Q8 (L194).** We replaced the symbol “↓’’ with “Proof in App. F’’ to improve clarity.
> - **Q10 (Tables 3 & 4).** Thank you for catching this. The parameter count for LaBraM-Base+iSICE was incorrect due to a formatting mistake. It has been corrected to 20.3K, consistent with other methods.
>
> All corresponding corrections have been incorporated into the revised manuscript and highlighted in blue.
>
>
> ## 3 (Q6). The Meaning of “Collapsing Spatiotemporal Tokens’’ in GCP.
>
> "The Collapsing" meaning is that standard GCP ignores the spatiotemporal relationships among tokens. Specifically, for a backbone output $X \in \mathbb{R}^{D \times T \times N}$, GCP first flattens the channel and temporal dimensions into $Z \in \mathbb{R}^{(D \cdot T) \times N}$, and then computes a covariance matrix
> $$
> \Sigma_{\mathrm{GCP}} = \frac{1}{N-1} \sum_{n=1}^N (z_n - \bar{z})(z_n - \bar{z})^\top \in \mathbb{R}^{N \times N},
> $$
> where $z_n$ are the token features，$\bar{z}$ is their mean. This “collapse’’ means that GCP models only the global covariance of features and do not retain token-wise spatiotemporal structure.
>
> Our method also produces one SPD descriptor, but through a different process: we first build per-token quotient Gaussian embeddings $Y_n$, then compute their Riemannian mean and covariance $(Y^m, Y^c)$, and finally embed them into a Riemannian Gauss embedding $G$. Therefore, this descriptor reflects statistics over token-level Gaussian embeddings, rather than the covariance of flattened features.
>
>
> ## 4 (Q7). Clarification of the Quotient Gaussian Space $\mathcal{QN}(n)$
>
> Your interpretation is correct and fully aligned with our intention. $\mathcal{QN}(n)$ denotes the quotient space of Gaussian distributions modulo positive diagonal scalings, and each element $[\Sigma,\mu]$ is indeed an equivalence class of Gaussians $(\Sigma',\mu)$ whose covariances differ only by diagonal rescaling and therefore share the same correlation matrix. Eq. (5) corresponds to the canonical representative of this class.
>
> We have rewritten the definition accordingly in the revised manuscript to explicitly present $\mathcal{QN}(n)$ as a quotient space in Eq. (5) and to clarify that $(C,\mu)$ denotes the correlation–mean representative of each equivalence class.

---

> ### Author Response · Authors · 2025-11-28
> **Response to Reviewer Pf1i (2/2)**
>
> ## 5 (Q9). The Relation Between Eq. (4) and Eq. (7)
> They correspond to the same operation. In the revision, we clarify this by using Eq. (4) to give the general definition of the Fréchet Mean (FM), and Eq. (7) to present its specific instantiation on the SPD manifold under the AIM. The relevant sentences have been rewritten to eliminate ambiguity.
>
> ## 6 (W1). Performance Improvements of RHOP
>
> We computed the absolute performance gains of RHOP over all compared pooling heads (BIOT/LaBraM backbones and all GCP variants: iSQRT-COV, SVD-Padé, and iSICE) across all datasets and metrics. Among these 92 improvements, **65 exceed +1%**, accounting for **70.7% of all comparisons**. This includes many substantial improvements, such as:
> * **+10.08%** (TUEV BIOT non-pretrained, SVD-Padé → RHOP, Balanced Acc.)
> * **+16.33%** (TUEV BIOT non-pretrained, iSQRT-COV → RHOP, Kappa)
>
> When comparing only against backbone models (i.e., BIOT vs BIOT+RHOP or LaBraM vs LaBraM+RHOP), RHOP still achieves consistently meaningful gains. Out of 24 backbone-level comparisons, **13 exceed +1%**, corresponding to **56.5% of all backbone improvements**. Representative examples include:
> * **+6.73%** on TUEV (BIOT non-pretrained, Balanced Acc.),
> * **+2.91%** on TUEV (BIOT pretrained, Balanced Acc.),
> * **+1.04%** on TUAB (LaBraM pretrained, Balanced Acc.),
>
> These results demonstrate that the gains of RHOP are far from marginal. They reflect consistent improvements across datasets, backbones, and evaluation metrics, confirming that RHOP provides meaningful advantages over both standard backbone pooling and advanced GCP-based methods.
>
> **References**
>
> [a] MAtt: A Manifold Attention Network for EEG Decoding.
>
> [b] SPD domain-specific batch normalization to crack interpretable unsupervised domain adaptation in EEG.
>
> [c] Multiclass brain–computer interface classification by Riemannian geometry.
>
> [d] Riemannian geometry for EEG-based brain-computer interfaces: a primer and a review.
>
> [e] Manifold-regression to predict from MEG/EEG brain signals without source modeling.
>
> [f] On the interpretation of linear Riemannian tangent space model parameters in M/EEG.
>
> [g] Is second-order information helpful for large-scale visual recognition?
>
> [h] Towards faster training of global covariance pooling networks by iterative matrix square root normalization.
>
> [i] Learning partial correlation based deep visual representation for image classification.

---

### Author Response · Authors · 2025-12-04
**General Response**

We sincerely thank the reviewers $\color{blue}{\rm Pf1i (R1)}$, $\color{red}{\rm VvTH (R2)}$ $\color{purple}{\rm sQZN (R3)}$ $\color{orange}{\rm 88CQ (R4)}$ and the AC for their valuable time and constructive feedback. Their comments greatly improved the clarity and completeness of the paper. We have carefully addressed all concerns raised during the rebuttal, and all corresponding revisions have been incorporated into the updated manuscript.

## Positive Feedback from Reviewers

* **Clear theoretical motivation**
  * $\color{blue}{\rm R1 }$. Clear theoretical motivation. Nice translation of the theory into a practical method.
  * $\color{red}{\rm R2}$. The method is conceptually interesting.
  * $\color{purple}{\rm R3}$. The work is conceptually interesting. The motivation is creative, and such cross-domain adaptation deserves attention.
  * $\color{orange}{\rm R4}$. The paper addresses a well-motivated and significant problem. The proposed solution, RHOP is built on a solid theoretical foundation.
* **Comprehensive experiments**
  * $\color{blue}{\rm R1}$. Comprehensive, compelling experiments.
  * $\color{orange}{\rm R4}$. Comprehensive and rigorous experimental validation. Diverse Benchmarks. Multiple Backbones. Multiple Training Paradigms. Strong Baseline Comparisons.
* **Well-written and clear presentation**
  * $\color{blue}{\rm R1}$. well written.
  * $\color{red}{\rm R2}$. The paper is well-written with clear methodology and good presentation.
  * $\color{orange}{\rm R4}$. The paper is well-written, well-structured, and easy to follow.
* **Strong performance and broad applicability**
  * $\color{orange}{\rm R4}$. The strong performance in the "from scratch" and "linear probing" settings is particularly impressive.

## Reviewer Concerns and Our Responses

1. **The general applicability of RHOP** ($\color{blue}{\rm R1},\color{purple}{\rm R3}$). We clarified that RHOP is motivated by EEG signal, but naturally extends to other multi-channel temporal modalities (ECG/MEG/fMRI). The corresponding discussion has been added to **App. H**.
2. **Missing backbone (CBraMod in ICLR'25)** ($\color{red}{\rm R2}$). We added experiments on CBraMod and showed RHOP consistently outperforms its original classifier and all GCP heads. The results and analysis have been included in **App. E.1**.
3. **Rank deficiency in covariance matrix** ($\color{red}{\rm R2}$). We detailed actual temporal-segment and channel dimensions and applied standard SPD regularization, ensuring all covariance matrices remain strictly SPD.
4. **Assessment of the Gaussianity assumption** ($\color{red}{\rm R2}$). We provided comprehensive Gaussianity analyses showing that temporal-segment features are near-Gaussian and stable, whereas GCP token features are heavy-tailed, validating RHOP’s modeling choice. The analysis has been added to **Sec. 5.3**.
5. **Marginal empirical gains** ($\color{blue}{\rm R1},\color{red}{\rm R2},\color{purple}{\rm R3}$). We summarized 52 comparisons where 75% exceed +1% improvements, indicating that RHOP provides consistent, meaningful performance benefits across models and datasets.
6. **Theoretical motivation using temporal-token covariance.** ($\color{purple}{\rm R3}$). We showed that temporal segments form the statistically appropriate unit for second-order modeling, unlike unstable token-level covariance.
7. **The detail experimental protocol** ($\color{purple}{\rm R3}$). We fully specified subject-independent/LOSO splits, preprocessing pipelines, repeated runs, and regularization to ensure reproducibility. Additional details have been included in **App. C**.
8. **The Ablation of the Fréchet Mean Iteration and (k,k')** ($\color{orange}{\rm R4}$). We showed that 1 FM iteration matches the performance of multiple iterations, and that moderate $(k,k')$ values provide the best balance between expressiveness and stability, supported by theoretical intuition for choosing $k$ or $k'$. The FM ablation has been added to **App. E.2**, and the $(k,k')$ analysis has been included in **Sec. 5.3** and **App. E.3**.
9. **Other writing issues** ($\color{blue}{\rm R1},\color{orange}{\rm R4}$). All notation, definitions, table inconsistencies, and unclear symbols were corrected for clarity.

We believe these updates and additions effectively address the reviewers’ concerns and strengthen the technical clarity and applicability of RHOP.

Thank you very much for your time and consideration!

Best regards,

Authors of Submission #2313

---

### Meta-Review · Area_Chair_87my · 2026-01-06

**Summary:**

This paper introduces Riemannian High-Order Pooling (RHOP), a plug-and-play classification head for EEG foundation models. The core idea is to replace standard pooling, or CLS-only heads, with a principled aggregation of first- and second-order token statistics on the SPD manifold, using quotient Gaussian embeddings and Riemannian Gaussian aggregation. Across multiple EEG benchmarks, backbones (BIOT, LaBraM, and CBraMod, added in the rebuttal), and training regimes (scratch, fine-tuning, and linear probing), RHOP delivers gains with minimal computational overhead.

The main weaknesses raised by the reviewers were:
-  VvTH: missing a key backbone baseline (CBraMod), potential non-SPD issues in covariance estimation, unjustified Gaussian assumption, and concerns that gains are marginal
- sQZN: physiological/theoretical motivation for temporal-segment covariance (vs the more standard spatial covariance), under-specified experimental protocols and dataset ambiguity (esp. BCIC2B), applicability beyond EEG
- 88CQ: clearer justification/ablation of design choices (single-iteration Fréchet mean approximation, sensitivity to k, k', and motivation for the specific Riemannian-Gaussian block embedding)
- Pf1i: broadly positive, mentioned modest gains and EEG-specific scope, along with several clarity/presentation issues.

Overall, reviewers found that the contribution is technically principled, empirically well supported after rebuttal, and addresses a gap in EEG foundation model pipelines with a broadly reusable, low-overhead head design.

**Reviewer Concerns:**

Most concerns were addressed convincingly by the rebuttal and the revisions:

- Missing baseline (VvTH):  added CBraMod experiments showing RHOP consistently improves over the original CBraMod classifier and over competing GCP heads.
- Rank-deficiency (VvTH): the authors clarified the dimensions used for covariance estimation, and add standard SPD regularization to ensure strict positive definiteness.
- Gaussianity assumption: (VvTH, sQZN): the authors added Gaussianity analyses showing temporal-segment features are close to Gaussian, whereas token-mixed GCP features are heavy-tailed, motivating why token-level covariance is statistically unstable.
- Marginal gains (VvTH, Pf1i): addressed with aggregated statistics over many comparisons and examples of significant improvements on key settings.
- Motivation for temporal-segment covariance + relation to prior Riemannian EEG literature (sQZN): more straightforward argument for why RHOP operates on semantic temporal-segment embeddings produced by the backbone (not raw EEG).
- Ablations and justification of design choices (88CQ): addressed with FM iteration ablations (showing saturation after a few steps), a clearer description of k, k' selection, and an explanation of why the Gaussian embedding is beneficial.

 Overall, the rebuttal resolves the majority of the reviewer's concerns.

**Reviewer Scores:**

-  VvTH: the main concerns are addressed, so it would perhaps become a 6. Only the "marginal gains" part is subjective, but in my view, the authors convincingly demonstrate the merits of the method.
- sQZN: the idea behind the method has been explained more clearly, and experimental details have been added. This reviewer might have bumped their score to a 6.
- 88CQ and Pf1i would have stayed 8, as their questions are convincingly addressed.

---

### Decision · Program_Chairs · 2026-01-26

Accept (Poster)